# Deep chlorophyll maximum and nutricline in the Mediterranean Sea: emerging properties from a multi-platform assimilated biogeochemical model experiment

Anna Teruzzi[1], Giorgio Bolzon[1], Laura Feudale[1], Gianpiero Cossarini[1]

[1]Istituto Nazionale di Oceanografia e di Geofisica Sperimentale - OGS, Trieste, 34100, Italy

*Correspondence to*: A. Teruzzi (ateruzzi@inogs.it)

**Abstract.** Data assimilation has led to advancements in biogeochemical modelling and scientific understanding of the ocean. The recent operational availability of data from BGC-Argo floats, which provide valuable insights into key vertical biogeochemical processes, stands to further improve biogeochemical modelling through assimilation schemes that include

float observations in addition to traditionally assimilated satellite data. In the present work, we demonstrate the feasibility of joint multi-platform assimilation in realistic biogeochemical applications by presenting the results of one-year simulations of Mediterranean Sea biogeochemistry. Different combinations of satellite chlorophyll data and BGC-Argo nitrate and chlorophyll data have been tested, and validation with respect to available independent non-assimilated and assimilated (before the assimilation) observations showed that assimilation of both satellite and float observations outperformed the assimilation

of platforms considered individually. Moreover, the assimilation of BGC-Argo data impacted the vertical structure of nutrients and phytoplankton in terms of deep chlorophyll maximum depth, intensity, and nutricline depth. The outcomes of the model simulation assimilating both satellite data and BGC-Argo data provide a consistent picture of the basin-wide differences in vertical features associated with summer stratified conditions, describing a relatively high variability between the western and eastern Mediterranean, with thinner and shallower but intense deep chlorophyll maxima associated with steeper and narrower

nutriclines in the western Mediterranean.

## 1 Introduction

In recent years, biogeochemical modelling has significantly contributed to the knowledge of key aspects of marine ecosystem processes at both local and global scales (Fennel et al., 2019). While quality assessment advancements have improved our confidence in model results (Hipsey et al., 2020), intrinsic limitations still exist because of unrepresented processes, uncertainty

in parameterization and numerical approximation (Dowd et al., 2014). On the other hand, emerging observation systems have provided valuable information on the biogeochemical state and processes in the ocean (Chai et al., 2020; Claustre et al., 2020; Groom et al., 2019; Muller-Karger et al., 2018; Roemmich et al., 2019). However, observations can be sparse and unevenly distributed in time and space (in situ), limited to the ocean surface (satellite remote sensing) and generally affected by

calibration and measurement errors (Bittig et al., 2019; Xing et al., 2020). Data assimilation (DA) aims to increase knowledge and representation of processes by integrating models with information obtained from observations.

In ocean biogeochemical modelling, the assimilation of satellite ocean-colour observations has been successfully applied in research and operational applications at both global and regional scales (Fennel et al., 2019; Groom et al., 2019). Chlorophyll concentration is the most commonly assimilated variable since the first applications of ocean biogeochemical DA (Ciavatta et al., 2016; Dorofeyev and Sukhikh, 2018; Ford and Barciela, 2017; Ford, 2020; Gehlen et al., 2015; Mattern et al., 2017; Pradhan et al., 2019; Ratheesh et al., 2016; Santana-Falcón et al., 2020; Song et al., 2016; Teruzzi et al., 2018; Tsiaras et al., 2017). However, assimilation of the ocean-colour diffuse attenuation coefficient, phytoplankton functional types, particulate organic carbon and inherent optical properties has been suggested as promising alternative to chlorophyll assimilation (Ciavatta et al., 2019, 2018, 2014; Dutkiewicz et al., 2019; Jones et al., 2016; Pradhan et al., 2020; Shulman et al., 2013; Skákala et al., 2018; Xiao and Friedrichs, 2014).

Ocean-colour observation assimilation takes advantage of the frequent, large-scale satellite observations related to the microbial biology of the upper ocean. However, the information includes the ocean surface layers only. In the context of data assimilation, the application of this information to deeper ocean layers requires approximations and assumptions. Vertical covariance must be parameterized by synthetic precalculated vertical profiles in variational schemes (Teruzzi et al., 2018), while EnKF-like (ensemble Kalman filter) schemes may have limitations in effectively impacting deeper ocean layers (Fontana et al., 2013; Hu et al., 2012). Indeed, some EnKF-like applications introduce limitation to the increments in subsurface layers through localization in the vertical direction to address spurious correlations (Goodliff et al., 2019; Pradhan et al., 2019).

In situ observations provide information on processes occurring in the ocean interior (e.g., deep chlorophyll maximum, vertical fluxes of nutrients or organic matter). An operational framework of biogeochemical observations has been recently introduced by BGC-Argo floats, which routinely deliver biogeochemical observations for the open sea (typically chlorophyll, oxygen, nitrate concentrations, optical properties and pH) with a profiling frequency ranging from 5 to 10 days (Bittig et al., 2019; Chai et al., 2020; Claustre et al., 2020; D'Ortenzio et al., 2020; Organelli et al., 2017). Data from BGC-Argo floats provide valuable insights into key vertical biogeochemical processes, such as the seasonal progression of stratified and mixed conditions and their impacts on the dynamics of phytoplankton and nutrients (e.g., Barbieux et al., 2019; D'Ortenzio et al., 2020, 2014; Fommervault et al., 2015; Lavigne et al., 2013; Mayot et al., 2017; Mignot et al., 2014). The first examples of the assimilation of float observations into biogeochemical models have improved the estimates of the vertical variability in biogeochemical variables (Cossarini et al., 2019; Verdy and Mazloff, 2017). Moreover, the potential benefits of integrating BGC-Argo observations with satellite data and biogeochemical modelling have been demonstrated by recent observing system simulation experiments (OSSEs) and parameter optimization studies (Ford, 2021; Germineaud et al., 2019; Wang et al., 2020).

In the present work, we demonstrate the feasibility of joint multi-platform assimilation in realistic biogeochemical applications by presenting the results of one-year simulations of Mediterranean Sea biogeochemistry using the MedBFM model system that includes the OGSTM transport model and the low trophic-level biogeochemical flux model BFM (Salon et al., 2019) offline coupled with the NEMO-OceanVar MFS model (Oddo et al., 2014, 2009). Different combinations of satellite

chlorophyll data and BGC-Argo nitrate and chlorophyll data have been assimilated using an upgraded version of the 3DVarBio variational assimilation scheme that was previously applied in single-platform assimilation (Cossarini et al., 2019; Teruzzi et al., 2019, 2018, 2014). The relatively high number of deployed BGC-Argo floats in the Mediterranean Sea and the specific seasonal variability in phytoplankton-nutrient dynamics and the noticeable west-east gradient of deep chlorophyll maximum depth of the Mediterranean Sea (DCM; Lavigne et al. 2013) make this basin a suitable location for the implementation and verification of a multi-platform assimilation system. The simulations carried out in the present work have been validated with respect to available assimilated and non-assimilated (before the assimilation) observations and have been investigated in terms of assimilation impact. Moreover, since a good simulation skill was demonstrated by the validation, the model outcomes have been used to explore the basin-wide differences in the vertical features associated with summer stratified conditions, when the DCM and nutricline are well established over the whole Mediterranean Sea (Barbieux et al., 2019; Lavigne et al., 2013; Lazzari et al., 2012; Mignot et al., 2014).

Section 2 describes the observation datasets used for the assimilation and the biogeochemical model and assimilation scheme setup. The results and discussion are provided in Sections 3 and 4, respectively.

## 2 Methods

The Mediterranean Sea biogeochemistry was simulated for one year(2015) with four different assimilation setups and a reference run without assimilation using the MedBFM model system that is operationally implemented in the Copernicus Marine Environment Monitoring Service (CMEMS) and provides nominal biogeochemical products for the Mediterranean Sea (Bolzon et al., 2020; Salon et al., 2019).

### 2.1 Observations

### 2.1.1 Satellite chlorophyll

The surface chlorophyll data used for assimilation included both open-sea and coastal observations (Teruzzi et al., 2018) and were obtained from the satellite multi-sensor product OCEANCOLOUR_MED_CHL_L3_NRT_OBSERVATIONS_009_040 (i.e., a merged product of MODIS-AQUA, NOAA20-VIIRS, NPP-VIIRS and Sentinel3A-OLCI sensors[1]). Original products, provided at a daily frequency and a horizontal spatial resolution of 1 km, were weekly averaged and interpolated at the model grid resolution to be used in the assimilation and validation following the strategy previously implemented in Teruzzi et al. (2014 and 2018). Two levels of quality check were performed on observations. The first quality check was made independently from data assimilation, and consisted in removing observations whose anomalies were higher than 3 times the daily climatology standard deviation in order to remove spikes. A second pre-data-assimilation check rejected satellite observations

---

[1] https://resources.marine.copernicus.eu/?option=com_csw&view=details&product_id=OCEANCOLOUR_MED_CHL_L3_NRT_OBSERVATIONS_009_040

not suitable for assimilation, excluding observations whose mismatch value with respect to the model was higher than 10 mg m$^{-3}$ to keep the range of model-observation differences consistent with the assumed uncertainties levels. The threshold was calibrated by a statistical analysis of the model-observations mismatches of the REF run. Through the pre-assimilation criterion, about 2% of satellite chlorophyll values were considered not suitable for the assimilation and rejected. With the described satellite chlorophyll data pre-processing, the yearly mean weekly coverage of surface chlorophyll was equal to 91% over the Mediterranean Sea with coverage higher than 95% in some sub-basins (alb, swm1, nwm, tyr1, ion1 and lev2; names of sub-basins in Fig. 1). As expected, the highest coverage occurred in summer (nearly 95% between May and August).

### 2.1.2 BGC-Argo floats

The BGC-Argo data used in the present study included vertical profiles of chlorophyll, nitrate and oxygen. The data were obtained from the Coriolis Data Assembly Centre and processed with the BGC-Argo community procedures (Bittig et al., 2019; Johnson et al., 2018; Schmechtig et al., 2018, 2015; Thierry et al., 2018). As for satellite observations, two levels of quality check were applied on float observations. In the first check, which excluded unrealistic values, an expert judgment procedure was applied, because the relative novelty of BGC-Argo data sets release may lead to a higher occurrence of poorly reliable observations. In particular, nitrate profiles were rejected if the surface value was higher than 3 mmol m$^{-3}$, chlorophyll profiles were checked for negative values (rejection of single negative observations), and quenching correction was performed by imposing a constant chlorophyll value in the mixed layer. The exclusion of nitrate values higher than 3 mmol m$^{-3}$ in the surface layers was based on the analysis of climatological values provided in the EMODnet dataset (Buga et al., 2018). In the pre-data-assimilation check (second quality check phase) observations were rejected when the mismatch with respect to the model was higher than 5 mg m$^{-3}$ and 2 mmol m$^{-3}$ for chlorophyll and nitrate, respectively. As for satellite, these values were based on a statistical analysis of the model-observations mismatches in REF. The pre-assimilation check excluded nearly 3% and less than 1% of the float nitrate and chlorophyll observations were excluded, respectively. The number of profiles used for assimilation and validation were 1484 for chlorophyll, 718 for nitrate and 794 for oxygen. Float profiles covered the whole Mediterranean Sea but with a larger sampling in the western regions than in the other regions (Fig. 1). In particular, the southern part of the Ionian Sea lacks float measurements. Since BGC-Argo floats are focused on and designed to operate in pelagic areas, shallow basins (northern Adriatic and Aegean Seas) were not sampled.

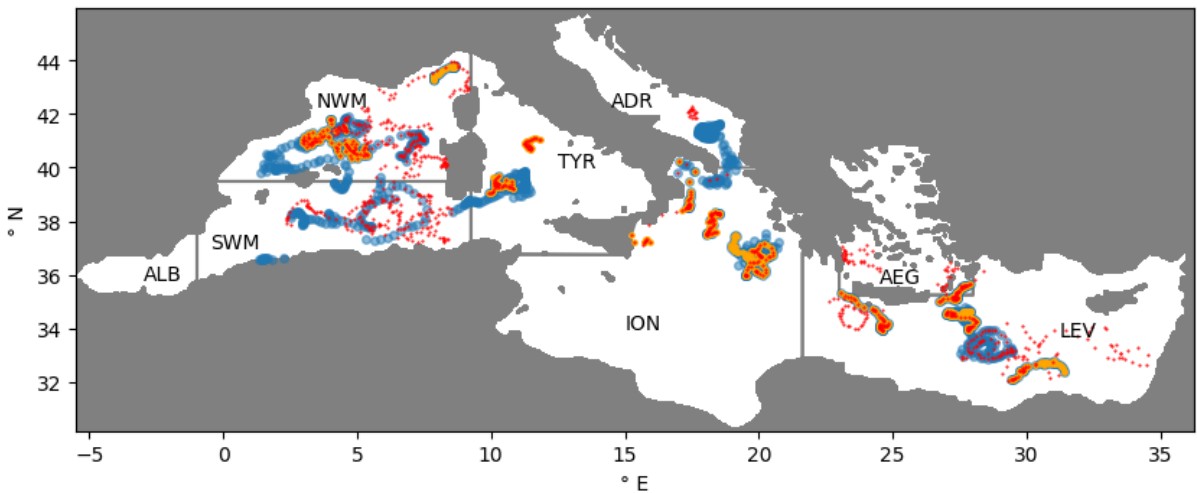

**Figure 1: Positions of BGC-Argo floats equipped with sensors to provide chlorophyll (blue), nitrate (orange) and oxygen (red) concentrations and limits of the subbasins: Alboran (ALB), southwestern Mediterranean (SWM) and northwestern Mediterrean (NWM), Tyrrhenian (TYR), Ionian (ION), Adriatic (ADR), Aegean (AEG) and Levantine (LEV) Seas.**

## 2.2 Model system


The MedBFM system used in this study is the biogeochemical component of the Mediterranean CMEMS model system aimed at providing short-term forecasts and multi-annual reanalysis, and consists of the coupled physical-biogeochemical OGSTM-BFM model and the 3DVarBio assimilation scheme (Salon et al., 2019). The OGSTM-BFM is designed with a transport model (OGSTM) and a biogeochemical reactor featuring the biogeochemical flux model (BFM). The OGSTM transport model, which

is a modified version of the OPA 8.1 transport model (Lazzari et al., 2010), was recently upgraded to resolve the free surface and variable volume-layer effects on the transport of tracers and is fully consistent with the off-line coupling of the NEMO3.2 version (Salon et al., 2019). The biogeochemical flux model (BFM) is a medium complexity low trophic-level ecosystem model designed to describe energy and material fluxes through both "classical food-chain" and "microbial food-web" pathways (Thingstad and Rassoulzadegan, 1995). This model includes nine plankton functional types (4 phytoplankton, 4 zooplankton

and one bacteria) and takes into account the co-occurring effects of multi-nutrient interactions (Lazzari et al., 2012). The BFM was recently developed with a carbonate system (Cossarini et al., 2015) and revised phytoplankton nutrient uptake processes (Lazzari et al., 2016). Moreover, the BFM has been implemented in the Mediterranean Sea in a wide range of applications, including 1D and 3D configurations aimed at studying the interaction between optics and biogeochemistry (Terzić et al., 2021, 2019), conducting climate scenario simulations (Lazzari et al., 2014), estimating carbon sequestration services (Melaku Canu

et al., 2015), simulating high-resolution coastal dynamics in marginal seas (Cossarini et al., 2017), analysing temporal scales of multi-decadal variability (Di Biagio et al., 2019), and studying CDOM (chromophoric-dissolved organic matter) spatiotemporal variability (Lazzari et al., 2021). 3DVarBio is the data assimilation scheme for the correction of phytoplankton

biomass and nutrient concentrations (i.e., nitrate and phosphate) using surface chlorophyll from satellite observations and vertical profiles of chlorophyll and nitrate from the BGC-Argo floats.


**Model setup**

In the current application, the MedBFM was forced offline with outputs from the NEMO3.2 model of the Mediterranean CMEMS model system (Simoncelli et al., 2016), which provides daily mean physical dynamics (i.e., horizontal and vertical current velocities, vertical eddy diffusivity, potential temperature, salinity, sea surface height in addition to surface data for

solar shortwave irradiance and wind stress). The mesh grid was based on a 1/16° longitudinal scale factor and a 1/16°cos(φ) latitudinal scale factor. The vertical mesh grid accounted for 70 vertical z-levels: 25 in the first 200 m of depth, 31 between 200 and 2000 m, and 14 below 2000 m.

Initial conditions were provided by a two-year spin-up simulation forced by 2015 physical fields in perpetual mode. The two-year spin-up was initialized by profiles of biogeochemical variables as provided by the EMODnet_int climatology, which

merges the in situ EMODnet data collections (Buga et al., 2018) and the datasets listed in Lazzari et al. (2016) and Cossarini et al. (2015). In the Atlantic area (i.e., west of the Strait of Gibraltar to the 9° longitude) tracer concentrations were relaxed to climatological seasonally varying profiles. Seasonal profiles of phosphate, nitrate, silicate, and dissolved oxygen were derived from an analysis of the climatological World Ocean Atlas 2018 data (Garcia et al., 2019) and the EMODnet_int dataset. The biogeochemical open boundary conditions at the Dardanelles Strait were obtained through a Dirichlet-type scheme that uses

climatological values from the literature (Salon et al., 2019). Atmospheric deposition rates of inorganic nitrogen and phosphorus were set according to the synthesis proposed by Ribera d'Alcalà et al. (2003), while terrestrial inputs of nutrients (nitrogen and phosphorous) from 39 rivers were obtained from the PERSEUS FP7-287600 project dataset (deliverable D4.6[2]). The terrestrial nutrient discharge rates were climatological (average of the 2000-2015 period) and took into account seasonal variability at a monthly scale based on varying monthly water discharge.

**2.3 3D variational assimilation**

In 3DVarBio, assimilation is performed through the minimization of a cost function that is defined on the basis of Bayes' theorem (Lorenc, 1986) as the weighted sum of the square mismatches between the model background state $x_b$ (the model state before the assimilation) and the analysis $x_a$ (the assimilation result) and the observations $y$. Each square mismatch is weighted according to its accuracy estimations, meaning that $x_a - x_b$ is weighted by the background error covariance matrix

**B** while $(y - H(xb))$ by the observation error covariance matrix **R**:

$$J(x_a) = (x_a - x_b)^T \mathbf{B}^{-1}(x_a - x_b) + (y - H(x_b))^T \mathbf{R}^{-1}(y - H(x_b)). \tag{1}$$

---

[2] http://www.perseus-net.eu/assets/media/PDF/deliverables/3321.6_Final.pdf

In eq. (1) $y - H(x_b)$ is usually named innovation and $H$ the observational operator that maps the values of the model background state $x_b$ in the observation space. In our application $H(x_b)$ are model values of the variables observed by satellite or floats at observation locations. Through the minimization of the cost function (1), the assimilation provides the analysis $x_a$, i.e., the optimal weighted distance from both $y$ and $x_b$. According to Weaver et al. (2003) and Dobricic and Pinardi (2008), the solution of the assimilation step (i.e., the increment $x_a - x_b$) is calculated by defining a control variable $v$ and a transformation matrix $V$ such that $x_a - x_b = Vv$ and $B = VV^T$. Moreover, the matrix $V$, which transforms increments from the control space to the model space, is decomposed into a sequence of operators that characterize different aspects of the error covariances: the vertical error covariance ($V_V$), the horizontal error covariance ($V_H$) and the biogeochemical state variable error covariance ($V_B$) (Teruzzi et al., 2014). Recent developments of the scheme include the upgrade of $V_H$ to become anisotropic to address the assimilation of satellite coastal observations (Teruzzi et al., 2018), the upgrade of $H$ and $V_V$ to address the assimilation of BGC-Argo float profiles (Cossarini et al., 2019), and the parallel recoding of the horizontal filter applied in $V_H$ (Teruzzi et al., 2019). In the present study, the 3DVarBio assimilation scheme was adapted to assimilate float nitrate dataand both satellite and float chlorophyll data and to provide corrections on all the phytoplankton variables and on nitrate and phosphate concentrations. In particular, in addition to the covariance between the assimilated chlorophyll and the 17 BFM variables describing the phytoplankton functional types (Teruzzi et al., 2014), the $V_B$ operator now includes the covariance between nitrate and phosphate calculated on a validated 20-year simulation of the MedBFM system (Teruzzi et al., 2016). Observation error covariance matrix $R$ has been assumed to be diagonal, and they are unchanged with respect to previous applications for satellite and float chlorophyll observations (Cossarini et al., 2019; Teruzzi et al., 2018), while a uniform and constant error has been assigned to float nitrate observations (0.24 mmol m$^{-3}$) according to triple-collocation error estimates (Mignot et al., 2019).

### 2.3.1 Assimilation setups

Provided that single-sensor chlorophyll data assimilation has already been demonstrated (Teruzzi et al., 2018; Cossarini et al., 2019), satellite (Sc) and float (Fc) chlorophyll data assimilation simulations in 2015, together with a reference simulation without assimilation (REF), were used as benchmarks to evaluate the relative improvement in the joint data assimilation simulations of float chlorophyll and nitrate (Fcn) observations and float nitrate and both float and satellite chlorophyll observations (ScFcn; Table 1). Oxygen profiles, available from the BGC-Argo floats, were not assimilated but were used as independent observations for validation.

The weekly averaged maps of satellite chlorophyll concentration (Sec. 2.1.1) were assimilated once per week, while the assimilation of available vertical in situ profiles of chlorophyll and nitrate concentrations (Sec. 2.1.2) was performed every day. When satellite and float observations were both available, the assimilation was performed separately with satellite information assimilation occurring first since the different data densities impacted the effectiveness of the multi-platform assimilation. Through the $V_B$ operator of 3DVarBio, the assimilation of chlorophyll data provided increments for the four phytoplankton functional group 17 state variables for chlorophyll data assimilation and for nitrate and phosphate for nitrate

data assimilation. The application of simultaneous increments of nitrate and phosphate was a key element in the Mediterranean

basin, where both nutrients can act as limiting factors of phytoplankton growth (Lazzari et al., 2016). Adopting a conservative

approach, other nutrients were not updated by DA since their less relevant role as limiting factor in the Mediterranean Sea.

| Name | Assimilated | Updated | Covariances in $V_B$ |
|---|---|---|---|
| REF | - | - | - |
| Sc | Sat CHL | Phyto | CHL-Phyto |
| Fc | Float CHL | | |
| Fcn | as Fc + Float NIT | Phyto, Nutrients | CHL-Phyto |
| ScFcn | as Fcn + Sat CHL | | NIT-PHO |

**Table 1: Assimilation setups and names assigned to the simulations. Assimilated variables can be satellite chlorophyll (Sat CHL), float chlorophyll (Float CHL) and float nitrate (Float NIT). DA updates can be applied to the 17 phytoplankton variables (Phyto) and to nitrate and phosphate concentrations (Nutrients). The covariance between biogeochemical variables applied by V_B is that**

**between chlorophyll and phytoplankton variables (CHL-Phyto) and between nitrate and phosphate (NIT-PHO).**

## 3 Results

### 3.1 Validation with observations

The performance of the different assimilation setups (Table 1) was evaluated by comparing available assimilated observations

with model outputs before the assimilation and non-assimilated observations with daily mean model outputs. The root mean

square difference (RMSD) in four macro sub-basins and at eight different layers (Fig. 1 and Table 2) was calculated for the

BGC-Argo observations and in two seasons for the surface satellite chlorophyll observations.

| Name | Depths | Validated variables |
|---|---|---|
| L1 | 0-10 m | Chlorophyll Nitrate Oxygen |
| L2 | 10-30 m | |
| L3 | 30-60 m | |
| L4 | 60-100 m | |
| L5 | 100-150 m | |
| L6 | 150-300 m | Nitrate Oxygen |
| L7 | 300-600 m | |
| L8 | 600-1000 m | |

**Table 2: Names, depths and variables used for the 8 layers employed in the validation.**

### 3.1.1 Validation with the BGC-Argo floats

Considering the BGC-Argo float chlorophyll data, the RMSD values of the REF simulation were lower in the eastern subbasins

than in the other subbasins, with values higher than 0.1 mg m$^{-3}$ only in the NWM subbasin (Fig. 2). In general, subsurface

layers had a higher RMSD than the surface with a west-east increase in the depth of the maximum RMSD, which matches the west-east gradient of the deep chlorophyll maximum (DCM) in the Mediterranean Sea. The assimilation of satellite chlorophyll (Sc) data alone did not clearly affect the model skill with respect to float chlorophyll data. Indeed, the RMSD values with respect to the REF decreased by almost 10% in the surface layers of ION and LEV and increased by almost 10-15% in TYR and in the sub-surface L4 layer of ION and LEV.

On the other hand, and as expected, the assimilation of float chlorophyll data significantly reduced the RMSD values with respect to chlorophyll in all sub-basins in the Fc, Fcn and ScFcn simulations. RMSD reductions occurred especially in layers with the highest RMSD in the REF simulation, ranging between -13% and -20%, with the highest values in the western sub-basins. Differences in RMSD reductions between the Fc and Fcn simulations were not appreciable, meaning that the assimilation of nitrate did not decrease the benefit of float chlorophyll data assimilation. The joint assimilation of satellite chlorophyll data and float chlorophyll data and nitrate data caused variations in the RMSD with respect to float chlorophyll data that was both positive and negative according to the effect on RMSD with satellite chlorophyll data assimilation alone. The resulting RMSDs were lower or slightly higher than the REF RMSDs everywhere, with the unique exception of layer L4 in LEV.

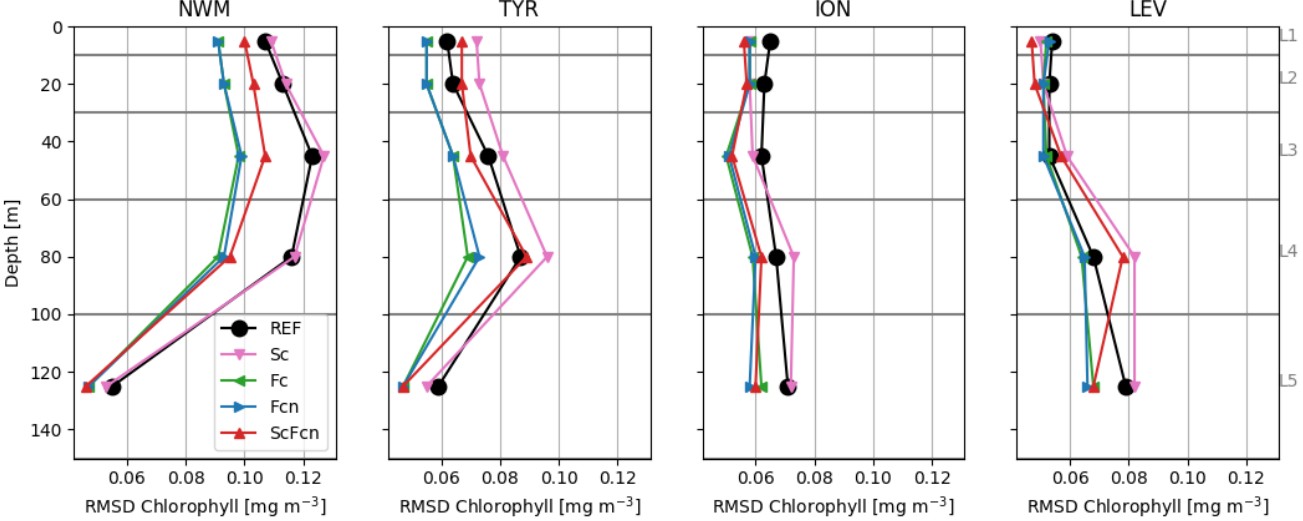

**Figure 2: RMSD between model simulations and BGC-Argo chlorophyll values in four sub-basins (Fig. 1). Grey lines indicate the limits of layers L1-L5 (Tab. 2) used to calculate the chlorophyll RMSD.**

Higher RMSD values with respect to float nitrate values (Fig. 3) were registered in the subsurface and deep layers (L5-L8) of the REF simulation in all the sub-basins except NWM. The RMSD values increased with depth following the vertical increase in nitrate concentration (Salon et al., 2019), while the RMSD values higher than 1 mmol m$^{-3}$ in the L1 and L2 layers in the NWM were related to an overestimation of surface nitrate during summer. Satellite data assimilation in the Sc simulation impacted nitrate indirectly through model dynamics after the DA increments on phytoplankton biomass, resulting in lower

RMSD values with respect to REF in the surface layers (L1-L4) in the NWM (-20%) and the opposite in the TYR (+20%). When float data were assimilated, a 5% reduction in nitrate RMSD occurred in the surface layers in the NWM and TYR in the Fc simulation, while a general reduction of up to 30% in several layers in the western sub-basins and up to 20% in ION and LEV were achieved with the assimilation of nitrate profiles in both the Fcn and ScFcn simulations.

Considering the joint assimilation in ScFcn, it is worth noting that an increase in the nitrate RMSD in L3 in the TYR resulted from the superimposition of the increase due to satellite chlorophyll data assimilation and the reduction due to float nitrate data assimilation. In general, the effects on RMSD with respect to nitrate in the joint satellite-float assimilation showed that the RMSD variations were almost additive: the RMSD reductions in the Sc simulation were reflected in more intense reductions in ScFcn than those in the Fcn simulation, while the opposite occurred in case of the RMSD increases in the Sc simulation.

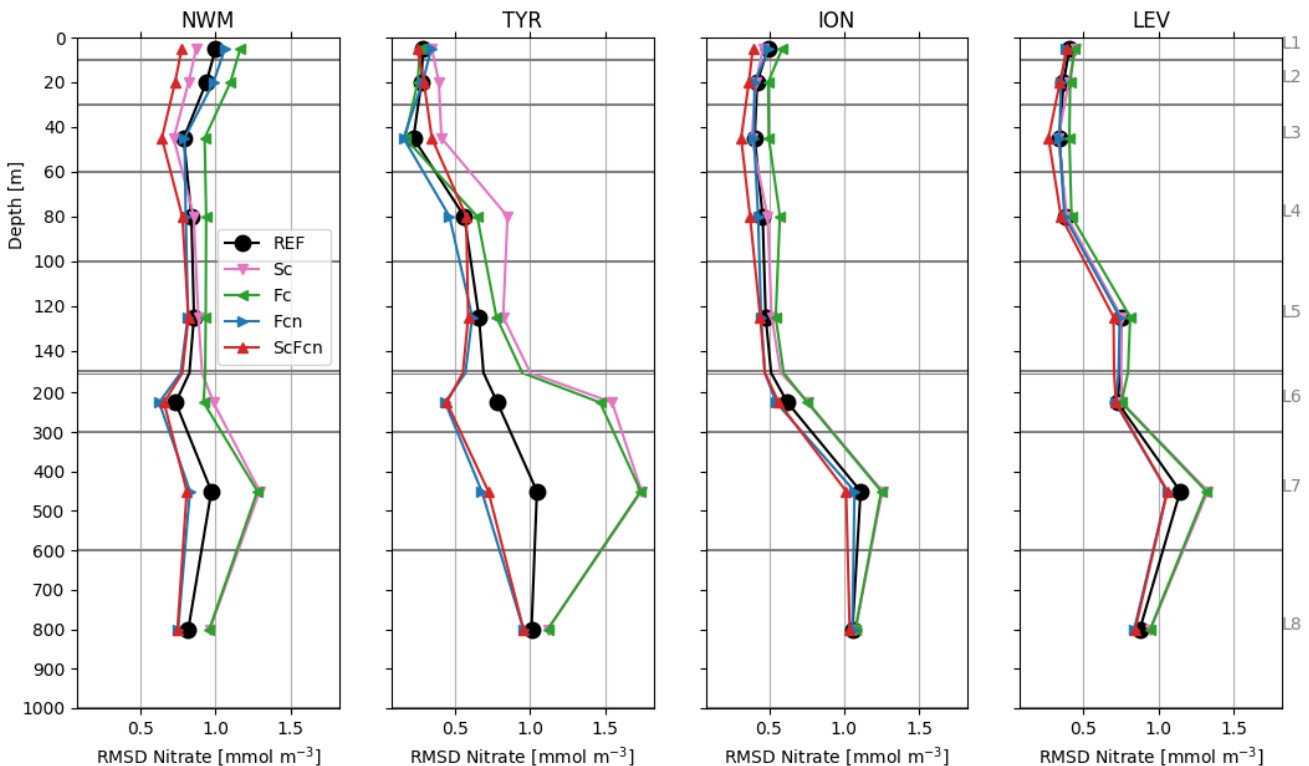

**Figure 3: RMSD between model simulations and BGC-Argo nitrate data in four sub-basins (Fig. 1). Grey lines indicate the limits of layers L1-L8 (Table 2) used to calculate the RMSD. The depth scale is different above and below 150 m (thick grey line).**

The RMSD between the float oxygen data and REF simulation (Fig. 4) increased from the surface (approximately 5-15 mmol m$^{-3}$) to the sub-surface and deeper layers (approximately 15-20 mmol m$^{-3}$) in the NWM and TYR sub-basins, while it was almost uniform along the vertical in the eastern sub-basins with ranges 2-10 mmol m$^{-3}$ and 7-17 mmol m$^{-3}$ in ION and

LEV, respectively. In particular, RMSDs very slightly differed among simulations, especially in the eastern sub-basins. In particular, the float assimilation had a very little effect on oxygen RMSDs and the effect of satellite chlorophyll assimilation is not univocal, since RMSDs are both slightly reduced (TYR) and increased (NWM) in Sc simulation. On the other hand, the float assimilation had a very little effect on oxygen RMSDs.

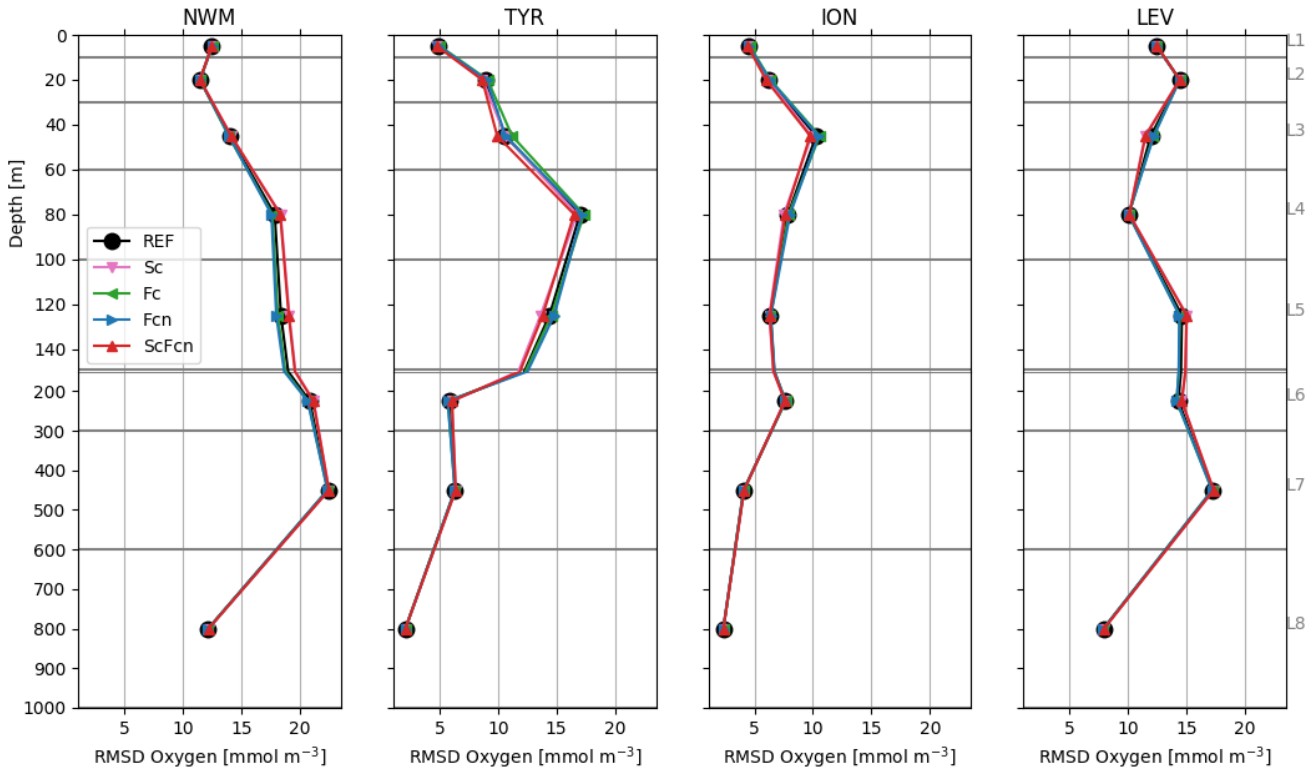

**Figure 4: RMSD between model simulations and BGC-Argo oxygen data in four sub-basins (Fig. 1). Grey lines indicate the limits of layers L1-L8 (Table 2) used to calculate the RMSD. The depth scale is different above and below 150 m (double grey line).**

### 3.1.2 Validation with satellite chlorophyll data

The RMSDs with respect to the satellite chlorophyll data (Fig. 5) in the REF simulation ranged between 0.07 and 0.11 mg m⁻³ in winter and between 0.013 and 0.035 mg m⁻³ in summer, consistently with the seasonal variation in the surface chlorophyll concentration in the Mediterranean Sea. Assimilation of float data alone had negligible effects on the RMSD calculated with satellite data (i.e., RMSD reductions between 0 and 3% in all the sub-basins except SWM in Fcn), whereas the assimilation of satellite chlorophyll data in the Sc and ScFcn simulations significantly reduced the satellite-chlorophyll RMSD with a gradient of reduction intensity from west to east. In fact, relative reductions were more intense than or close to 50% in ION and LEV and almost equal to 20% and 30% in SWM in winter and summer, respectively.

The skill performance analysis (Fig. 2-5) clearly showed that joint assimilation of satellite chlorophyll data and float chlorophyll and nitrate data (ScFcn simulation) significantly reduced the RMSD with respect to all the assimilated variables and did not induced any degradation of float oxygen observations, outperforming all the simulations with single-stream assimilation.

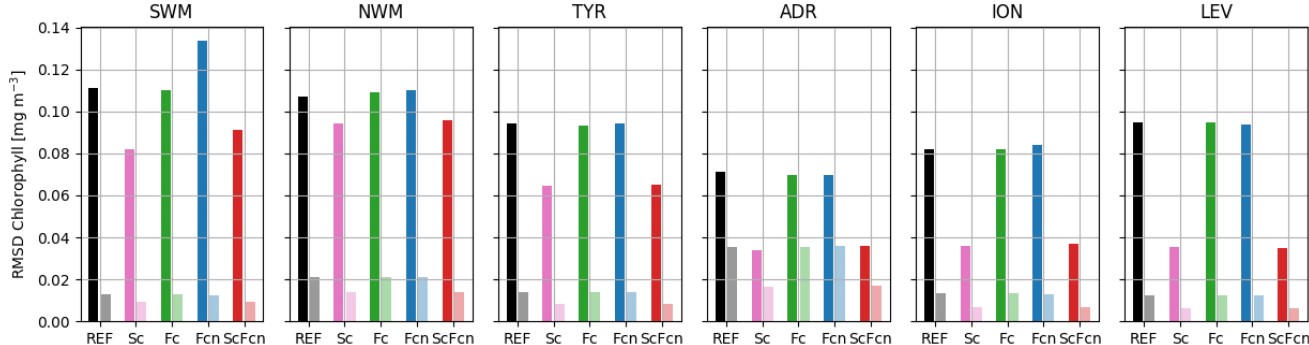

**Figure 5: RMSD between model simulations and satellite chlorophyll observations in six sub-basins (Fig. 1) for winter (left and darker bars) and summer (right and lighter bars).**

### 3.2 Data assimilation impact

Several factors (e.g., the number and position of available observations, the innovation values, the spatial covariances and those between the biogeochemical variables) influence the spatial extension and temporal persistence of the assimilation impact on biogeochemical dynamics. Several methods have been tested to measure the impact of observations on models, including conducting OSSE experiments (e.g., Ford, 2021; Germineaud et al., 2019) and introducing data impact indicators (e.g., Raicich and Rampazzo, 2003). To assess the impact of satellite and BGC-Argo float observations, the following indicator was evaluated:

$$I_{xy}(t) = \frac{\overline{|ScFcn(t) - REF(t)|_{200}}}{\overline{REF(t)_{200}}} \tag{2}$$

where $ScFcn(t)$ and $REF(t)$ indicate results at date $t$ of the ScFcn and REF simulations, respectively, and $|S(t) - R(t)|$ is their absolute difference. The subscript 200 represents the integral over the 0-200 m layer, while the overbar represents the average over the whole Mediterranean and over seasonal periods. The impact indicator $I_{xy}(t)$ was calculated for each assimilation date and each grid point, and then, statistically analysed and summarized on a seasonal base. The indicator $I_{xy}(t)$ quantifies how much an assimilated run deviates from the REF simulation, thus it is higher where and when the simulation deviates from REF, and is closer than REF to the assimilated observations. The seasonal median of the impact index values masked scattered assimilations (in particular in summer for chlorophyll and in winter for nitrate; Figs. 6 and 7) showing that satellite data, given their high observational density, almost always had a relevant impact over the whole basin. On the other hand, the 95[th] percentile of the $I_{xy}(t)$ distribution allowed us to highlight the areas where at least one assimilation remarkably ameliorated the model mismatch.

In almost the whole (i.e., 97%) Mediterranean Sea the $I_{xy}(t)$ 95th percentile for chlorophyll concentration was greater than 0.3 in winter (Fig. 6), while in summer, it was greater than 0.3 in 26% of the Mediterranean Sea and mainly in areas surrounding the float trajectories (Fig. 6, bottom map). The impact indicator calculated for the single data-stream assimilation runs (Fcn and Sc, not shown) confirmed that summer local DA effects were mainly due to float assimilation, while the relevant impact of ScFcn over the whole basin in winter was mainly related to satellite chlorophyll data assimilation.

Chlorophyll profiles averaged over the areas with $I_{xy}(t)$ 95th percentiles higher than 0.3 for REF and ScFcn revealed that DA impacts were different in winter and summer (Fig. 6, left panels). In winter, when impacts were mainly related to satellite observations, assimilation generally induced chlorophyll concentration reductions that were vertically homogeneous in the surface layer and then vanished almost at the bottom of the euphotic layer. On the other hand, the float-related assimilation impacts of summer were vertically localized around DCM and affected its depth and intensity.

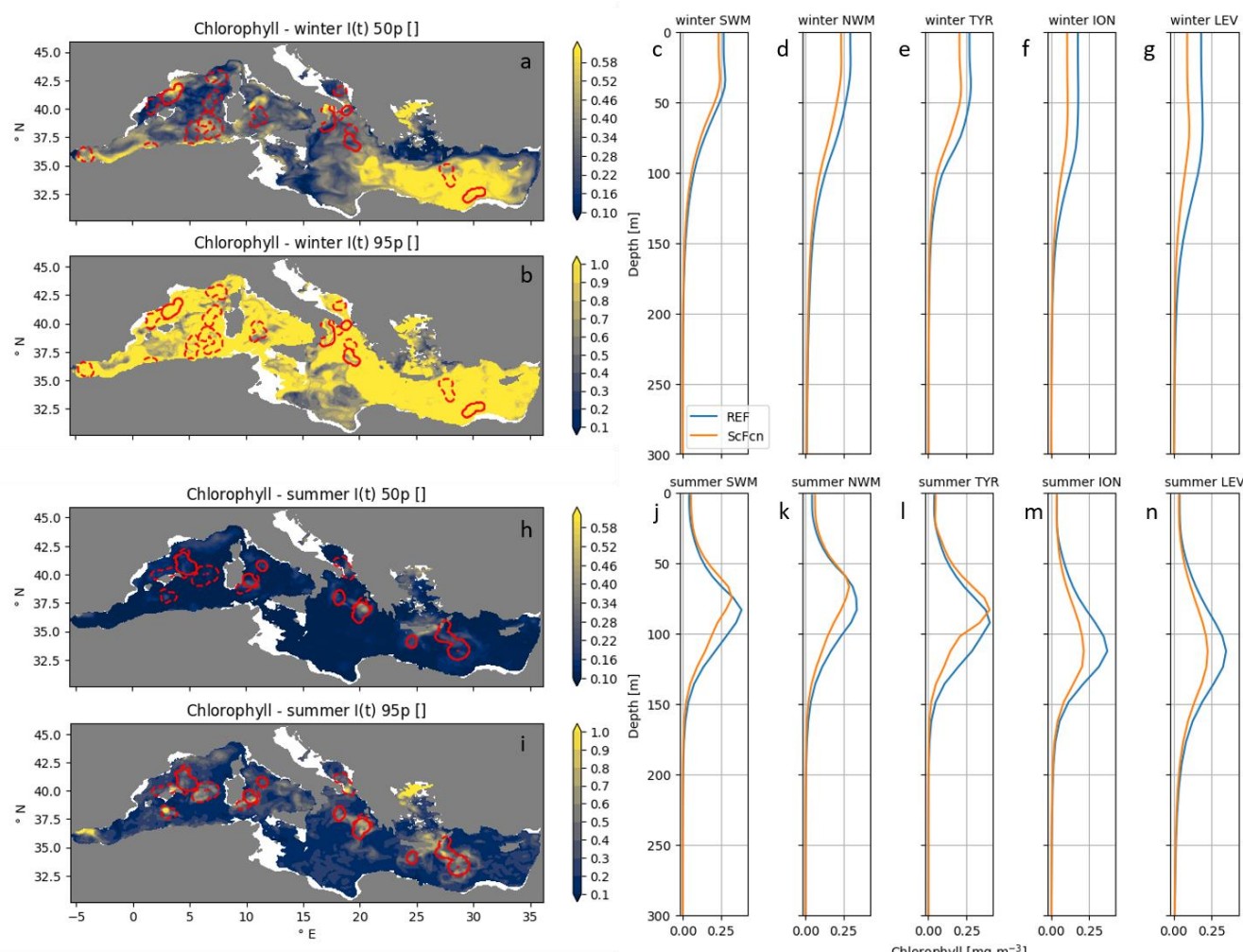

Figure 6: Maps of $I_{xy}(t)$ 50th and 95th percentiles for chlorophyll in winter and summer (left column, a, b and h, i panels, respectively); red contours identify the areas within 3 correlation radii from the float positions (dashed lines for floats without nitrate observations). Mean profiles at time and location with $I_{xy}(t)$ higher than 0.3 in five sub-basins for the ScFcn and REF simulations (five right panels) in winter (c-g) and summer (j-n).

The data assimilation impact on the nitrate 3D field was less intense than that for chlorophyll, and relatively high values of $I(t)$ were located mainly in float trajectory convolution areas (Fig. 7 maps). Satellite and float chlorophyll data in both single-stream and joint assimilation (not shown) had relatively low impacts on nitrate, and local patches of large assimilation impacts occurred mainly where float nitrate data were assimilated.

Analysing the seasonal distribution of the spatial data impact, the $I_{xy}(t)$ 95th percentile was higher than 0.3 only in the 3% of the Mediterranean Sea in winter, while the percentage increased to 12% in summer. Moreover, in summer, the $I_{xy}(t)$ 95th percentile was slightly higher in the basin outside the areas affected by float observations. The seasonal differences in the $I_{xy}(t)$ spatial distribution were due to two concurring effects: the lower number of floats equipped with nitrate sensors in

winter and the persistence of nitrate changes, as indicated by the enhancement in $I_{xy}(t)$ spatial homogeneity over time (not shown).

Data assimilation impacts on nitrate profile shapes (Fig. 5) were nearly homogeneous along the vertical with negligible effects

on nitracline depth and slope in ION in winter (positive ScFcn-REF differences) and in NWM in winter and in ION and LEV in summer (negative ScFcn-REF differences). In contrast, nitracline depth and slope were significantly affected by vertically non-uniform assimilation impacts (e.g., NWM and TYR in summer). Below 300 m (not shown), differences between REF and ScFcn propagate nearly constantly until 500 m and then tend to vanish between 500 and 600 m.

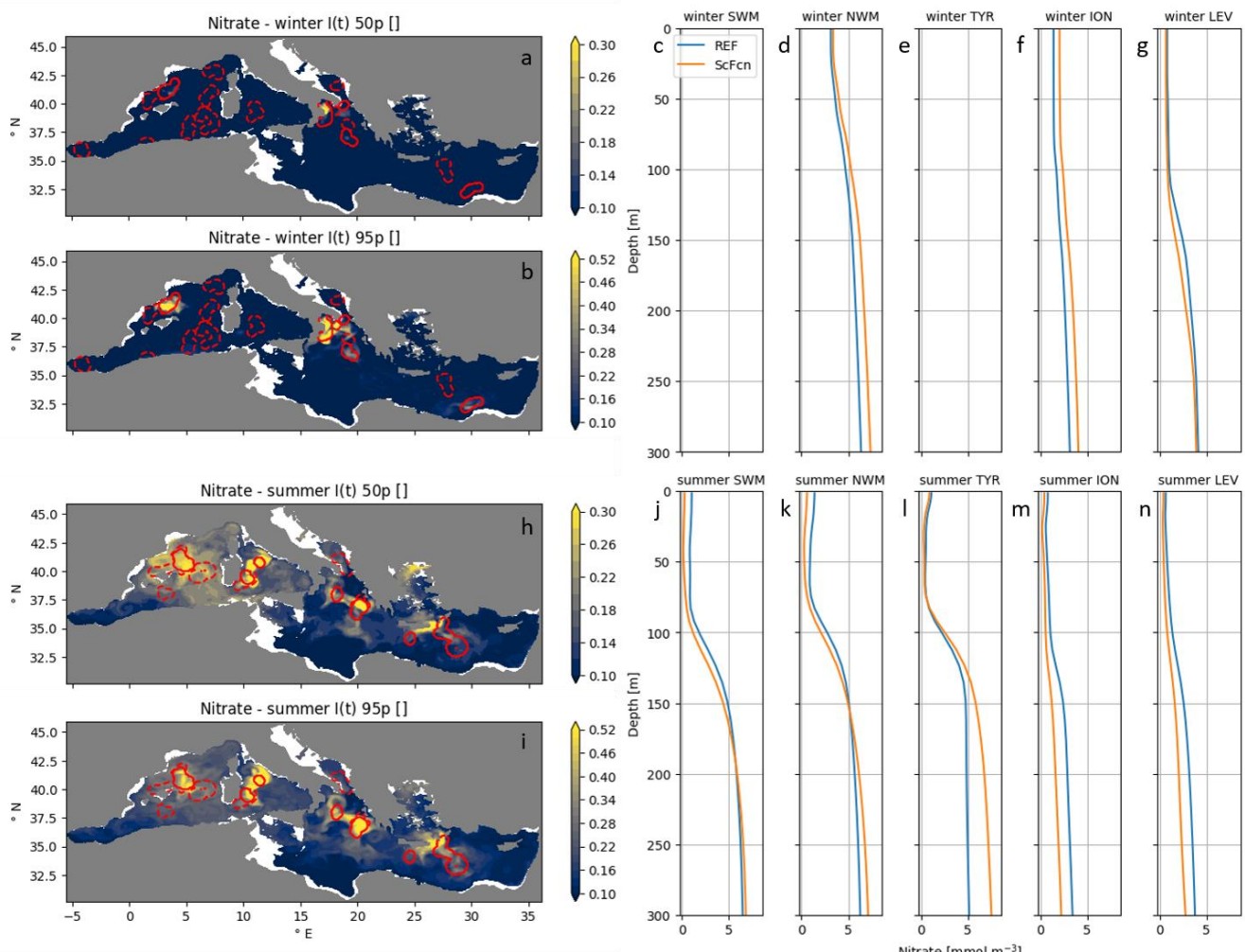

**Fig. 7. Maps of the $I_{xy}(t)$ 50$^{th}$ and 95$^{th}$ percentiles for nitrate in winter and summer (left column, a, b and h, i, respectively); red contours identify the areas within 3 correlation radii from the float position (dashed lines for floats without nitrate observations). Mean profiles at time and location with $I_{xy}(t)$ higher than 0.3 in five sub-basins for the ScFcn and REF simulations (five right panels) in winter (c-g) and summer (j-n).**

### 3.3 Nitracline and deep chlorophyll maximum

The good skill of the analysis simulation that integrates satellite and BGC-Argo observations to reproduce 3D fields of phytoplankton and nutrients (ScFcn simulation), supported the use of the simulation outputs to investigate some basin-wide biogeochemical features in summer, when processes in the vertical direction are dominated by stratified conditions. In particular, deep chlorophyll maximum (DCM) and nutricline dynamics were investigated by calculating several indexes that identified the vertical displacement and the profile shape (Table 5) and provided the representation of their west-to-east

variability in the Mediterranean Sea during summer (Figs. 8 and 9).

| | Quantity [unit] | Definition |
|---|---|---|
| **DCM** | Depth [m] | Depth of the maximum chlorophyll concentration. |
| | Intensity [mg chl/m$^3$] | Chlorophyll concentration at the DCM. |
| | Thickness [m] | Thickness of the layer where the chlorophyll concentration is higher than half the difference between chlorophyll at the DCM and chlorophyll at the surface. |
| **Nitracline** | Depth [m] | Depth where the maximum first derivative of the nitrate concentration along the vertical is located (with the exclusion of maxima at depths lower than 30 m). |
| | Slope [mmol N/m$^4$] | Thickness of the layer with the first derivative of the nitrate concentration is larger than 75% of the derivative at the nitracline. |
| | Thickness [m] | Mean of the first derivative of the nitrate profile in the layer used to define nitracline thickness. |
| **Light availability** | PAR at DCM [mol quanta/m$^2$/d] | Photosynthetic available radiation calculated at the DCM. |

**Table. 5: Indexes evaluated in simulation ScFcn during summer stratification.**

Consistent with previous findings (Lavigne et al., 2013; Lazzari et al., 2012; Mignot et al., 2014), the simulated DCM depth exhibited a west-east gradient (Fig. 8) ranging from nearly 80 m in the western basins to values higher than 100 m in the eastern basins. Moreover, in the ScFcn simulation, the DCM and DBM (deep phytoplankton biomass maximum) depths mostly

coincided with a correlation coefficient higher than 0.9. The DCM thickness and the chlorophyll concentration at the DCM also showed spatial gradients over the Mediterranean Sea (Fig. 9): moving eastward, as the DCM deepens, the DCM thickness increases while its intensity (chlorophyll concentration at DCM) decreases.

According to the paradigm that the DBM is a layer where both light and nutrients are co-limiting factors for phytoplankton (Cullen, 2015), the DCM and nitracline depths were spatially correlated in the Mediterranean Sea (Fig. 9). Indeed, similar to

the DCM depth, the nitracline depth showed a west-to-east gradient, and given the proposed definitions (Table 5), the nitracline was located slightly above the DCM, with a nearly constant 15 m gap. As shown for the DCM, moving eastward, the nitracline thickness varied in the same way as its depth. Indeed, the nitracline thickness was higher in the eastern (50 m) than in the western sub-basins (30 m, Fig. 9). The nitracline-slope gradient exhibited an opposite sign moving eastward, with values ranging between 0.08-0.12 mmol m$^{-4}$ and 0.02-0.5 mmol m$^{-4}$ in the western and eastern Mediterranean, respectively (Fig. 9).

Even if the Mediterranean Sea is a small marginal sea, it exhibits a wide range of summer vertical conditions: a 25% thinner and shallower but intense DCM associated with a more than double steeper and 33% narrower nitracline in the western Mediterranean than in the more oligotrophic eastern Mediterranean.

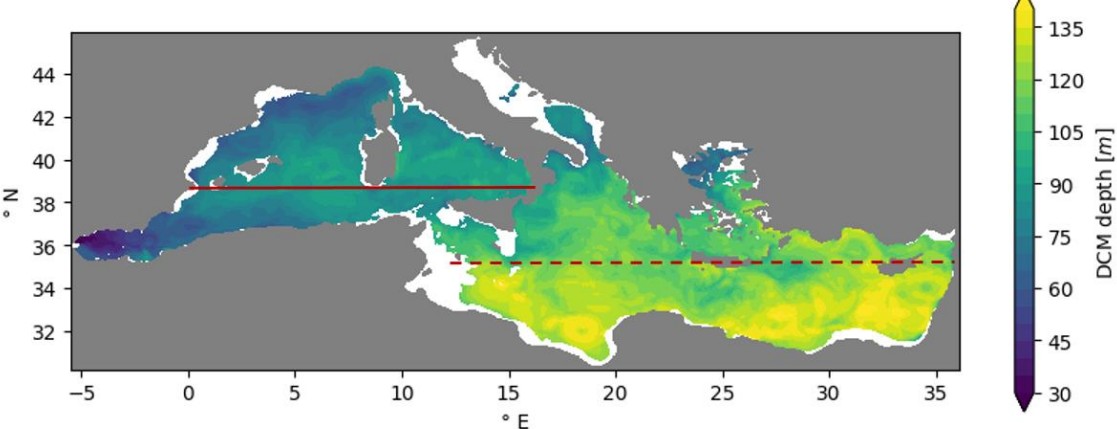

**Figure 8: Mean DCM depth [m] calculated over summer in areas with water depths higher than 200 m. Western and eastern**
**meridional averages of Fig. 9 are calculated along red continuous and dashed lines, respectively, excluding the Adriatic (ADR) and Aegean (AEG) Seas.**

The simulated values of PAR at the DCM depth (Fig. 9) ranged between 1.5 and 2 mol quanta m$^{-2}$ d$^{-1}$ moving westward from the Sicily Channel and between 0.6 and 1 mol quanta m$^{-2}$ d$^{-1}$ in the eastern Mediterranean, with higher PAR-at-DCM values occurring between 24°E and 26°E only. The west to east decreasing values of PAR at DCM is documented in a study based
on BGC-Argo observation (Mignot et al., 2014), even if the reported values were slightly lower than in Fig. 9.

The DBM can be considered a nutrient trap layer, where all the nutrient fluxes from the bottom layer are consumed by phytoplankton. Thus, since it is confined upward by nutrient depletion and downward by the absence of light, higher irradiance at the DBM (equivalent to DCM in our case) can indicate a higher rate of nutrient uptake by phytoplankton and related higher production (Cullen, 2015). In our results, this hypothesis was confirmed by higher nutrient uptakes in the western sub-basins
than in the eastern sub-basins at the DCM depth: the nitrate and phosphate uptake ranges were equal to 1.0-2.3 10$^{-2}$ mmol m$^{-3}$d$^{-1}$ and 1.0-2.1 10$^{-3}$ mmol m$^{-3}$ d$^{-1}$ in the western Mediterranean, while in the eastern Mediterranean, they decreased to 0.5-1.2 10$^{-2}$ mmol m$^{-3}$d$^{-1}$ and 0.4-0.6 10$^{-3}$ mmol m$^{-3}$d$^{-1}$, respectively. The spatial variability in nutrient uptake was correlated with the phytoplankton primary production gradient simulated in the DCM layer. In fact, the simulated primary production

maximum in the DCM layer was equal on average to 4 mgC m$^{-3}$ d$^{-1}$ in the western Mediterranean and to 2.5 mgC m$^{-3}$ d$^{-1}$ in the eastern Mediterranean (a detailed primary production scenario simulated by the same BFM model with a very similar setup is presented in Fig. 2.2.2 of Schuckmann et al., 2020). The higher rate of biological activity produced a sharper transition from the surface depleted zone to the deeper nutrient-rich zone in the western Mediterranean. Additionally, the higher nitrate concentration in the mesopelagic layer contributed to increasing the nitracline steepness in the western sub-basins (Fig. 7) and hence the vertical supply of nutrients and higher nutrient consumption and production in the DBM layer.


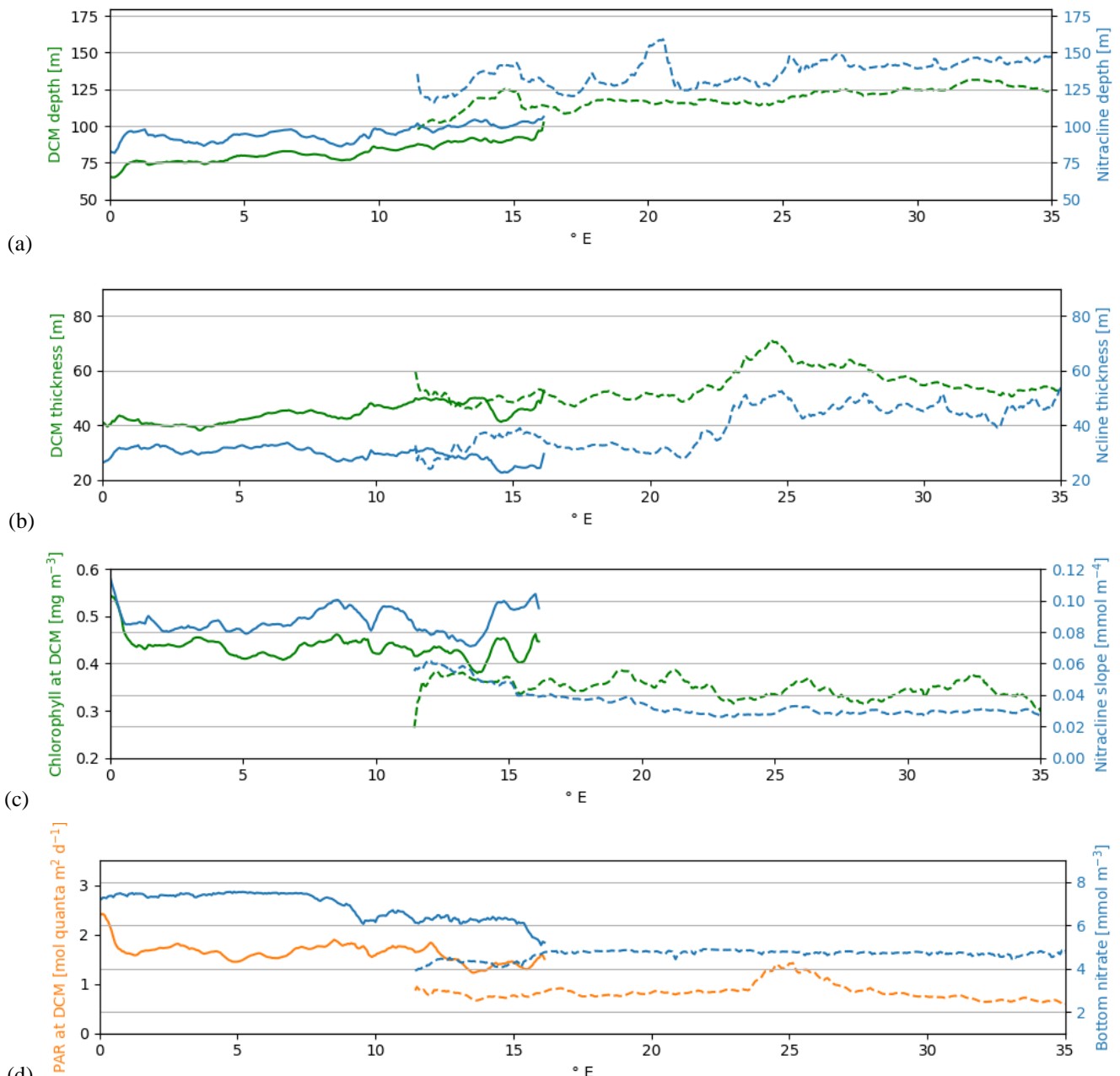

(a)

(b)

(c)

(d)

**Figure 9: Meridional averages of summer water column indexes (Table 5) along red lines in Fig. 8 for western (continuous line) and eastern (dashed line) sub-basins: DCM (green) and nitracline (blue) depth [m] (a); DCM (green) and nitracline (blue) thickness [m] (b); DCM intensity (green) as chlorophyll concentration [mg m$^{-3}$] and nitracline slope (blue) [mmol m$^{-4}$] (c); PAR at DCM depth (orange) [mol quanta m$^{-2}$ d$^{-1}$] and bottom nitrate concentration (blue) [mmol m$^{-3}$] (d).**

## 4 Discussion

The results of the present work demonstrate the feasibility of assimilating multi-stream biogeochemical observations in real ecosystem simulations, showing the potential and the impact of emerging observation systems such as the BGC-Argo network. Moreover, using assimilation setups that differently combine satellite chlorophyll and BGC-Argo float chlorophyll and nitrate observations, the reciprocal effects of each observation source showed that the impact of two or more data streams resulted in an almost superimposition of the assimilation effects on the simulation results. Even if the significantly different observed

densities of the two data streams considered here (i.e., much higher in the satellite data than in the float data) could lead to the suppression of the impact of the less populated data stream, it is also true that the satellite surface observations and float profiles demonstrated to act mostly in different seasons (winter versus summer, respectively) and at different pelagic layers, leading to an additive superimposition of single assimilation effects. These results were obtained by adopting a sequence of assimilation events for the different data streams and letting the model dynamically adjust the increments. In particular, the

assimilation frequency was set to weekly and daily for satellites and floats, respectively, in accordance with previous studies on single-stream assimilation (Cossarini et al., 2019; Teruzzi et al., 2018, 2014). However, alternative strategies can mitigate the effects of higher observation densities excessively reducing the impact of the other more sparse and more scarce observations. For instance, an error in the spatial covariance component can be included in the observation error covariance matrix (Moore et al., 2019) to take into consideration the spatial correlation of neighbouring high-density observations (e.g.,

satellite observations). Keeping the observation error covariance matrix diagonal, another option can be to increase satellite observation errors close to the locations of the assimilated float profiles. Either considering the off-diagonal element of the observation error matrix or dynamically adjusting its diagonal elements close to float positions requires considerable work to revise the formulation scheme and tune the errors.

On the other hand, as shown in a recent OSSE experiment (Ford et al 2020), the availability of a large and homogeneous

float coverage can generate a full-domain impact of float assimilation. The foreseen increase in the BGC-Argo network (Bittig et al., 2019; Claustre et al., 2020) will move in that direction. However, the need to elongate life floats will probably result in a decrease in the sampling cycle frequency from 5 to 10 days (Roemmich et al., 2019). Moreover, the potential effectiveness of the present BGC-Argo missions in covering the Mediterranean bioregions has been demonstrated at least for chlorophyll (D'Ortenzio et al., 2020). Thus, while it is desirable to increase the number of nitrate sensors, it must be

acknowledged that the density of BGC-Argo could not increase indefinitely in the future. In the Mediterranean Sea, a season-dependent sampling frequency that is higher during winter and spring surface blooms and lower during summer slow-dynamic conditions could compensate for the battery-saving needs and the maximization of float impact in joint float-satellite assimilation. In addition, to increase the spatial impact of data assimilation, future operational implementations can be based on pseudo profiles reconstructed by neural network approach such as CANYON-MED that uses the larger coverage

of the Argo network and oxygen sensors (Fourrier et al., 2020).

The present results of the joint assimilation of multi-stream chlorophyll data showed a mitigation of the increase in the RMSD of chlorophyll with respect to the non-assimilated chlorophyll dataset in the single-stream assimilation simulations (Figs. 2 and 5). The RMSD increase with respect to the non-assimilated chlorophyll data in the single-stream assimilation can be ascribed to discrepancies between the two datasets obtained by different measurement methods. Provided that florescence-
derived methods and reflectance-based models have different sensitivity and calibration patterns whose investigation is out of the scope of the present work, it is relevant to highlight that discrepancies have a seasonal distribution: higher-than-float satellite values occur in summer, and the opposite occurs in winter (Fig. 10). As a consequence of these discrepancies and of the propagation of information through the prescribed DA vertical covariance, the change in chlorophyll profiles due to satellite assimilation either reduced and increased the distance between modelled chlorophyll and BGC-Argo chlorophyll profiles
(Fig. 2). On the other hand, considering RMSDs with respect to satellite chlorophyll (Fig. 5), the effect of inconsistency between satellite and float chlorophyll was highlighted by the slight increase in RMSDs in simulations with float chlorophyll assimilation. In perspective, when the inconsistency between satellite and float will be solved, the multi-platform assimilation will provide improvements over large areas thanks to the relevant spatial coverage of satellite observations. In the meantime, the model, acting as a dynamical filter, integrates both sources of information together with the vertical covariance $\mathbf{V_V}$ operator
implemented in the assimilation scheme. Indeed, the vertical covariance is a key element that allows to integrate surface with sub-surface information. In our case the $\mathbf{V_V}$ operator was derived at a monthly scale from the simulation results averaged over sub-basins (Teruzzi et al., 2018). However, by adopting a higher temporal and spatial resolution for the definition of $\mathbf{V_V}$, local information (single satellite observation and float profile) can be better integrated trough the assimilation. Indeed, in physical oceanography, point-to-point varying $\mathbf{V_V}$ or a combination of seasonal pre-calculated and flow-dependent parts have been
successfully tested in assimilation applications (Dobricic et al., 2015; Storto et al., 2018; Storto and Oddo, 2019).

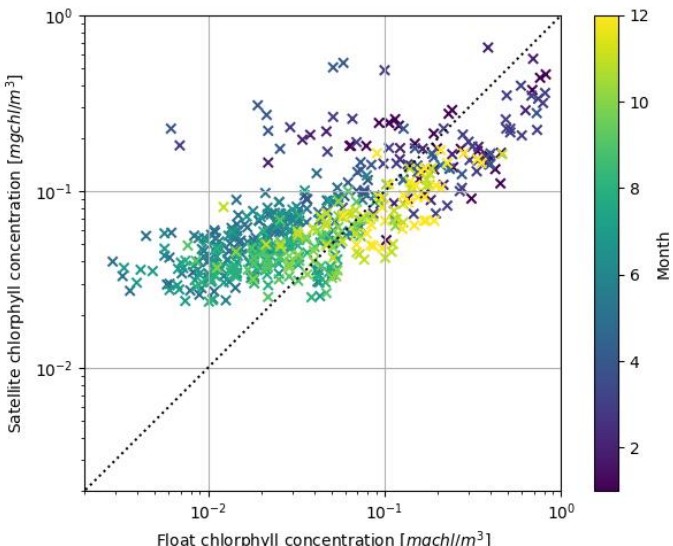

**Figure 10: Scatter plot on a logarithmic scale of float and satellite observations of chlorophyll concentration.**

The different assimilation setups investigated in the present work showed the reciprocal effects of the assimilation of different data streams and how they combined in the multi-stream assimilation. Considering effects on nitrate, the assimilation of
satellite chlorophyll reduced the nitrate RMSDs (computed on BGC-Argo data) with respect to REF simulation in all the sub-basins with exception of TYR (Fig. 3). The rather persistent and broad DA increments during late winter and early spring were acting to reduce overestimation of the bloom maxima, resulting in nearly uniform reductions of the phytoplankton biomass in the euphotic layer. In turn, by reacting to the new phytoplankton concentration, less nitrogen was eventually re-mineralized to nitrate. Thus, the nitrate concentration was modified in the direction of reducing the REF nitrate overestimation in the upper
layer. An analogous mechanism has been investigated in previous applications of satellite chlorophyll assimilation in the MedBFM system (Teruzzi et al., 2018, 2014). In the case of BGC-Argo chlorophyll assimilation, the changes in the phytoplankton profiles were less uniform (Cossarini et al., 2019), often alternating positive and negative increments along the same profile. It follows that the effects on nitrate profiles (through new uptake or release after mortality and exudation) were non-linear and not uniform, with relatively small impacts on the overall nitrate RMSDs. Concerning the assimilation effects
on the non-assimilated oxygen variable, the non-degradation of the oxygen RMSDs was a positive aspect of the assimilated simulations, while the limited and non-univocal effects of the assimilation on oxygen RMSDs were related to the interaction of a number of trophic processes (e.g., phytoplankton production and respiration, zooplankton and bacteria respiration) after the assimilation increments on phytoplankton biomass. Effects on non-assimilated biogeochemical variables are discussed in a number of works (e.g., Ciavatta et al., 2014; Ford, 2020; Mattern et al., 2017; Santana-Falcón et al., 2020; Simon et al., 2015;
Teruzzi et al., 2018; Tsiaras et al., 2017; Yu et al., 2018), where the non-degradation of non-assimilated variables is considered already a good result of the assimilation process. Moreover, the model-assimilation system acts as a filter so that, even if the

performance of the multi-platform assimilation is lower than anyone of the single assimilation, it produces a balanced solution with respect to all the available information.

The $V_B$ operator (biogeochemical covariance operator) plays a key role on the accuracy of results of the assimilated and non-assimilated variables. In this work, $V_B$ was based on the actual physiological status at each grid point for the phytoplankton components, while the nitrate-phosphate covariance was pre-calculated using a multi-annual simulation, and varied monthly and at sub-basin scale. Our results suggest that the relevant variability in the DCM and nutricline conditions across the Mediterranean Sea would require a $V_B$ including finer temporal and spatial scales covariances between phytoplankton and nutrients. An available option to extend the number of variables considered in $V_B$ and to increase its capability to represent covariance at finer scales consist of Kalman filter-based methods, among which recently emerging hybrid schemes (Carrassi et al., 2018) could be particularly relevant since allowing the combination of pre-calculated covariance with the one estimated by an ensemble Kalman filter approach. In the present application the covariance operator $V_B$ provided impacts on all the phytoplankton variables together with those on two nutrients (phosphate and nitrate), which can act as limiting factors of phytoplankton growth in the Mediterranean Sea (Lazzari et al., 2016). In perspective $V_B$ could be developed to include other variables, however, considering silicate, it should be noted that in OGST-BFM applications in the Mediterranean Sea silicate limitation is less relevant compared to nitrate or phosphate.

Recent literature highlighted the relevant role of assimilating vertical observations from BGC-Argo to improve the simulation of key biogeochemical processes (Ford, 2021; Germineaud et al., 2019; Wang et al., 2020). Our analysis of the assimilation impact showed that the description of several biogeochemical features of the euphotic layer (such as the DCM depth and intensity and nutricline depth) benefited from BGC-Argo chlorophyll and nitrate data. Furthermore, the results of the simulation that integrated float and satellite observations provided a validated three-dimensional description of Mediterranean Sea biogeochemistry. In particular, we investigated the DCM layer during summer-stratified conditions. The results were quantitatively consistent with previous estimations of DCM-depth over the Mediterranean Sea (Lavigne et al., 2013; Lazzari et al., 2012; Mignot et al., 2014) and qualitatively with the results of studies that investigated the variability in the main DCM and nutricline features according to different nutrient and light availability regimes (e.g., Aksnes et al., 2007; Barbieux et al., 2019; Beckmann and Hense, 2007; Cossarini et al., 2019; Cullen, 2015; Gong et al., 2017; Terzić et al., 2019). In particular, we showed that the DCM is shallower, more intense and less thick and occurs at higher light intensity with higher nutrient uptake by phytoplankton in the western Mediterranean than in the eastern Mediterranean. Correspondently, the nitracline and the phosphocline (not shown) are shallower, steeper and narrower. Moving eastward, the DCM and nutricline features change by as much as 50% (Fig. 9), which indicates that the Mediterranean Sea has relatively variable conditions despite being a small semienclosed basin (Bethoux et al., 1999; Malanotte-Rizzoli et al., 2014; Schroeder et al., 2016). Several factors contribute to the east-west differences in the DCM and nutricline properties. For instance, a higher light extinction factor and higher nutrient concentrations in the bottom layer than in the other layers have been associated with shallower DCM (or DBM) and nutricline depths and a steeper nutricline (Aksnes et al., 2007; Beckmann and Hense, 2007; Gong et al., 2017; Mellard et al., 2011; Mignot et al., 2014). Moreover, winter deep-mixing events can create conditions for a shallower DCM since a larger

availability of nutrients in the subsurface layer is made available by winter mixing in the northwestern Mediterranean (Mignot et al., 2014). In our simulations, both the higher bottom-layer nutrients and typical strong-mixing events in the western Mediterranean were consistently reproduced and sustained the east-west gradient of DCM depth. As in our simulation results, BGC-Argo observations in the Mediterranean Sea (Barbieux et al., 2019; Fommervault et al., 2015) showed a shallower, more intense and narrower DCM and a shallower and steeper nitracline in the western Mediterranean but nitracline depths closer to the DCM in the eastern Mediterranean, in contrast with the almost constant difference between the DCM and nitracline depths observed in our simulation. However, this discrepancy can be ascribed to the different methods used to define a nitracline. According to Salon et al. (2019), we defined nitracline as the layer where the slope was larger, corresponding to the interface between high and depleted nutrient concentrations. In fact, results more similar to those of Barbieux et al. (2019) have been obtained applying the same definition of nitracline depth to our simulation results (not shown). In a previous study based on BGC-Argo observations, the relation between the DCM and DBM in the Mediterranean Sea was also investigated (Barbieux et al., 2019; Mignot et al., 2014). As an emergent property of the BFM formulation and simulation setups, our results show that the subsurface depths of chlorophyll and biomass maxima coincided (similar to the findings of Mignot et al., 2014). On the other hand, the results of Barbieux et al. (2019) highlight that the two depths can be non-coincident under oligotrophic conditions typical of the eastern Mediterranean basins.

In addition to confirming previous findings on the spatial variability in the DCM layer characteristics in the Mediterranean Sea, the assimilated simulation presented in this work provides a full 3D and time-varying description of a number of biogeochemical variables, allowing to further investigate potential relations between the vertical distributions of phytoplankton, nutrients, light availability, and other physical forcings. The use of the results of assimilated simulations that include along-depth observations can integrate investigations on the DCM and its dynamics that to date have been based on observations, which can be sparse and not evenly distributed in time (Navarro and Ruiz, 2013; Ricour et al., 2021).

## 5 Conclusions

In this work, we presented the results of a set of simulations of Mediterranean Sea biogeochemistry that integrated BGC-Argo chlorophyll and nitrate observations and satellite chlorophyll observations using different assimilation setups. The results show that the assimilation of all the data streams outperformed the single-source assimilation when validated with respect to available observations, indicating that the assimilation of BGC-Argo observations has relevant (even if local) impacts on the vertical structure of nutrients and phytoplankton. The impacts of multi-variate profile assimilation are directly linked to the sampling frequency and dimension of the BGC-Argo network, which should increase to match the consolidated importance and relevance of satellite observation assimilation. Thus, in a perspective view, the multi-platform assimilation can improve model representation of both large-scale (hundreds to thousands of kilometres) to mesoscale features and be beneficial for robust reconstruction in global and regional reanalysis. The results of the simulation with multi-platform assimilation provided a 3D description of the basin-wide gradient of DCM and nitracline dynamics through specifically developed metrics (e.g., DCM

depth and intensity , and nitracline depth and steepness) and highlighted the role played by light and nutrient availability in the western and eastern Mediterranean Sea. Even if the Mediterranean Sea is a small marginal sea, it exhibits a wide range of

summer vertical conditions of the DCM and nutricline with thinner and shallower but intense DCM associated with a steeper and narrower nutricline in the western Mediterranean, while the opposite occurs in the eastern Mediterranean Sea.

**Code and data availability**

The BFM biogeochemical model and its documentation can be downloaded at the following address: http://bfm-community.eu/ (last access 8th April 2021). The variational assimilation code has been published in Teruzzi et al. (2019) along with the relevant

git repository (https://github.com/inogs/3dVarBio). The quality-controlled databases used in the present paper are publicly available from the SEANOE (SEA scieNtific Open data Edition) publisher at https://doi.org/10.17882/42182 (Argo, 2021).

**Author contribution**

AT and GC conceived the study. GB and AT developed the model code and performed the simulations. LF and GB prepared the observation datasets. AT conducted the analysis of the simulation results and wrote the first manuscripts draft. GC and AT

discussed and reviewed the manuscript. LF and GC participated to the final manuscript editing.

**Competing interests**

The authors declare that they have no conflict of interest.

**Acknowledgments**

This study has been conducted using EU Copernicus Marine Service Information.

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
