# Peer review of "Deep chlorophyll maximum and nutricline in the Mediterranean Sea: emerging properties from a multi-platform assimilated biogeochemical model experiment"

_Biogeosciences, 2021_

## Author Comment (AC1)

**Review by Sarah Schlunegger**

Referee comments on "Deep chlorophyll maximum and nutricline in the Mediterranean Sea: emerging properties from a multi-platform assimilated biogeochemical model experiment" by Teruzzi et al., currently in discussion at Biogeosciences.

Teruzzi et al presents the results from a data-assimilating biogeochemical model of the Mediterranean Sea. This work provides a novel case-study of the dual use of remotely-sensed (ocean color) and in situ (BGC-Argo) biogeochemical constraints in reconstruction of one year of the biogeochemical state of the Mediterranean Sea. Firstly, the data-informed model solution demonstrates fidelity with observations for a number of surface and depth-resolved biogeochemical metrics, such as the vertical position and longitudinal gradients of the deep chlorophyll maximum and its co-variance with the nutricline. Secondly, the study presents compelling evidence for the synergistic benefits of assimilating estimates of remotely-sensed chlorophyll concentrations in tandem with in situ, depth-resolved estimates of chlorophyll. Thirdly, the study discusses some of the nuanced differences between the impacts of assimilating satellite versus BGC-Argo chlorophyll upon the model solution. The most striking difference discussed is the strong seasonal signatures the different observational streams have upon the solution, with remotely-sensed chlorophyll providing stronger constraint during winter months, and in situ chlorophyll providing stronger constraint during summer months. Finally, implications for optimized sampling strategies, such as the recommendation that BGC-Argo increase sampling frequencies during the 'influential' summer season, are discussed.

This work represents a timely contribution to the underway community efforts towards optimizing the design of a global, autonomous biogeochemical observing network for use in constraining reconstructions of the evolving ocean state. I recommend this manuscript for publication after a few minor clarifications and elaborations are incorporated, as I outline below.

*We thank Sarah Schlunegger for the positive and constructive comments on the manuscript. All the points raised in her review are addressed below with our replies in green italic.*

Points of clarification:

The model setup describes the biogeochemical model as being coupled offline to the dynamical model. Coupled implies two-directional influence, i.e. that the biogeochemistry is both impacted by and impacts the dynamical fields. Is this the case?

If it is the case that the model set up is "coupled" in the true sense of the word, then the impact on the dynamical fields should also be presented.

If it is not the case, and in fact the biogeochemical model or assimilation of biogeochemical fields does not feed-back or modulate the underlying dynamical fields, then a more appropriate term to use is "forced" or "driven" – i.e. Line 135 should read "the MedBFM was forced or driven offline with output from MENO3.4-OceanVar…"

*In MedBFM, the biogeochemical variables are tracers advected and diffused by ocean dynamics and transformed by biogeochemical processes, adopting a source splitting operator (Butenschön et al., 2012) without feed-back on the dynamical fields. In particular, transport and advection are driven by precomputed daily 3D fields of currents and diffusivity, while T and S are used in the biogeochemical module. We will modify the manuscript following the Reviewer's suggestion to better explain this aspect.*

To improve accessibility, an additional paragraph of description or context of Eq. 1 should be included. This would involve, for instance, explanation of the "innovation" term (Line 159), and discussing the significance or intuitive purpose of the different covariance vectors.

*In the new version of the manuscript we will add a clarification on the terms used in eq. (1):*

*"In 3DVarBio, assimilation is performed through the minimization of a cost function that is defined on the basis of Bayes' theorem (Lorenc, 1986) as a weighted sum of the square of the mismatches between the model background state $x_b$ (the model state before the assimilation) and the analysis $x_a$ (the assimilation result) and the observations $y$. Each square mismatch is weighted according to the respective accuracy estimation, meaning that $x_a - x_b$ is weighted by the background error covariance matrix $B$ while $(y - H(x_b))$ by the observation error covariance matrix $R$:*

$$J(x_a) = (x_a - x_b)^T B^{-1}(x_a - x_b) + (y - H(x_b))^T R^{-1}(y - H(x_b)). \qquad (1)$$

*In eq. (1) the term $y - H(x_b)$ is usually named innovation and $H$ is the observational operator that provides the values of the model background state $x_b$ in the observation space. In our application $H(x_b)$ are model concentrations of the variables observed by satellite or floats at observation locations. Through the minimization of the cost function (1), the assimilation provides the analysis $x_a$ that minimizes its weighted distance from both y and $x_b$."*

Regarding the "Impact indicator" metric:

(a)      It needs to be explained that the impact indicator given by Eq 2, although it does not contain an explicit 'directionality' of the impact (i.e. towards better or worse agreement with the data-constraints), that the assimilation methodology works to push the model solution towards the data, therefore any non-zero value of I(t) represents a nominal improvement in model-data fit. From the equation alone, and without sufficient prior understanding of the methodology, this is not obvious, making it difficult to interpret if the additional streams of assimilated data are merely influencing or in fact improving the solution.

*Thank you for the very constructive comments about the Impact Indicator. Concerning the first comment (a), it is true that the impact is in the direction of the assimilated observations for grid-points corresponding to observation locations. On the other hand, grid-points without observations are corrected consistently with the solution of the DA scheme (e.g., the effects of surface chlorophyll assimilation on the water column profiles) with impacts vanishing for grid-points far from any observation locations. Thus, the impact indicator is higher where the assimilation brings the simulation toward the observations and this can be considered an improvement of the simulation with respect to nearby assimilated observation. As suggested by the Reviewer, we will modify the description of the impact indicator in order to highlight this aspect:*

*"[...] The subscript 200 represents the integral over the 0-200 m layer, while the overbar represents the average over the whole Mediterranean and over seasonal periods. The impact indicator I(t) was calculated for each assimilation date and each grid point, then, statistically analysed and summarized on a seasonal base. The indicator I(t) quantifies how much an assimilated run deviates from the REF simulation, thus it is higher where and when a simulation is closer than REF to the assimilated observations."*

(b) The maps of Fig. 6 and Fig. 7 present a very derived / abstract metric. These figures present the 50th and 90th percentile 'impact' of assimilated the given field during the given season. Interpretability of this abstracted metric would improve if an additional subpanel was included in each figure which showed a representative distribution of the impact-indicator for a single grid-cell (or averaged over a region), with markings at the 50th and 90th percentile. This could also provide opportunity to contrast the summer vs. winter distributions of the indicator. For instance, for chlorophyll, the winter distribution of the indicator

would be shifted toward "1" while the summer distribution would be closer to "zero". See a mock-up below. This will help orient the reader as to what the maps are presenting.

*We thank the Reviewer for this constructive comment about the Impact Indicator. Following the Reviewer's suggestion, we investigated the distribution of the Impact Indicator over winter and summer. In particular, we analysed frequency distributions of the impact indicator at some locations, and we provide in Fig. R1 two examples. The histograms confirm results shown in Fig. 6 and 7, with higher Impact factor percentiles in winter with respect to summer for chlorophyll and the opposite for nitrate (since the impact of sparse float observations is negligible at the simulation beginning). Further, the histograms are quite scattered since they depend on a number of factors (e.g., the number of observations in the area and assimilation changes during subsequent dates). Therefore, we think that the non-parametric statistics (percentiles) shown in the manuscript are a sound choice to quantify the impact of the assimilation on model solution and we prefer to avoid to insert the grid-point histograms.*

[Figure]

*Fig. R1. Frequency histograms of the Impact Indicator at two locations for chlorophyll (top) and nitrate (bottom) in winter (blue) and summer (orange). Vertical lines indicates the values of the 50th (p50, dashed line) and 95th (p95, continuous line) percentiles.*

The conclusions section would benefit from a final sentence that poses the significance of the study within the context of future advances in biogeochemical data assimilation within basin and global domains, i.e. something like " The multi-platform assimilation yielded improvements in model representation of largescale (hundreds to thousands of kilometers) bio-dynamical features and is suggestive of the applicability of this advancement to reconstructions of other ocean regions and the global domain."

*We thank the reviewer for this suggestion that helps to enlarge the perspective of our manuscript. We will add a comment about the applicability of multi-platform DA to other domains:*

*"The impacts of multi-variate profile assimilation are directly linked to the sampling frequency and dimension of the BGC-Argo network, which should increase to match the consolidated importance and relevance of satellite observation assimilation. Thus, in a perspective view, the multi-platform assimilation can improve model representation of both large-scale (hundreds to thousands of kilometres) to mesoscale features and be beneficial for robust reconstruction in global and regional reanalysis."*

Minor editorial and stylistic suggestions:

L7-10: Rewrite to something like: "Data assimilation has lead to advancements in biogeochemical modelling and scientific understanding of the ocean. The recent operational availability of data from BGC-Argo floats, which provide valuable insights into key vertical biogeochemical processes, stands to further improve biogeochemical modelling through assimilation schemes that include observations from floats in addition to traditionally assimilated satellite data." (bold is new)

*Thank you for the suggestion. We will include it in the revised manuscript.*

L16: "maximum depth, intensity and nutricline depth" (added a comma and removed the first 'and')

*We will change the sentence accordingly.*

L39: Add the following reference, which also provides motivaiton for the direct use of optical properties in data assimilation:

Dutkiewicz, Stephanie, Anna E. Hickman, Oliver Jahn, Stephanie Henson, Claudie Beaulieu, and Erwan Monier. 2019. "Ocean Colour Signature of Climate Change." Nature Communications. 10 (1). https://doi.org/10.1038/s41467-019-08457-x.

*We will inserted the reference.*

L40: Ocean-colour observation assimilation takes advantage of the frequent, large-scale satellite observations of ocean properties related to the microbial biology of the upper ocean.

L42: "deeper ocean layers requires approximations and assumptions."

*We will change the manuscript according to the two previous Reviewer's suggestions.*

L45: "localization" – what does this mean?

*In Ensemble Kalman filter (EnKF)-like data assimilation, "localization" is a method applied to limit the impact of the assimilation (increments) to areas relatively close to the observations. In particular, localization is applied when knowledge of spatial covariance is relevantly approximated due to the limited ensemble size. In this sense, localization avoid to insert spurious and non-realistic effects and can be applied not only to limit the spatial covariance but also on covariance among variables or on the time dimension. Concerning the use of "localization" in the sentence at L. 45, we are referring to vertical localization applied in some DA application of satellite ocean colour, meaning that the assimilation impacts are limited to a portion of the*

*water column through localization. We will modify the sentence as follows to make it more clear about "localization" in EnKF-like data assimilation:*

*"Vertical covariance must be parameterized by synthetic pre-calculated vertical profiles in variational schemes (Teruzzi et al., 2018), while EnKF-like (ensemble Kalman filter) schemes may have limitations in effectively impacting deeper ocean layers (Fontana et al., 2013; Hu et al., 2012). Indeed, some EnKF-like applications introduce limitation to the increments in subsurface layers through localization in the vertical direction to address spurious correlations (Goodliff et al., 2019; Pradhan et al., 2019)."*

L66: "ideal" – replace with "suitable" or "appropriate" – as it could be easily argued that a region with less coastal margins and more open-ocean conditons, where ocean color is more reliable, would be a more 'ideal' location to do a multi-platform assimilation.

*We will replace "ideal" with "suitable".*

L348: "In our results, this hypothesis was supported by higher nutrient uptakes in the western…"

*We will replace "scenario" with "hypothesis".*

L409: Remind readers of what Vv means, i.e. " Vv (vertical co-varience error)"

L415: Remind readers of what Vb means, i.e. " Vb (biogeochemical co-varience error)"

*We will add a remind of what Vv and Vb mean into brackets as suggested by the Reviewer.*

*References*

Butenschön, M., Zavatarelli, M., Vichi, M., 2012. Sensitivity of a marine coupled physical biogeochemical model to time resolution, integration scheme and time splitting method. Ocean Model. 52–53, 36–53. https://doi.org/10.1016/j.ocemod.2012.04.008

Goodliff, M., Bruening, T., Schwichtenberg, F., Li, X., Lindenthal, A., Lorkowski, I., Nerger, L., 2019. Temperature assimilation into a coastal ocean-biogeochemical model: assessment of weakly and strongly coupled data assimilation. Ocean Dyn. https://doi.org/10.1007/s10236-019-01299-7

Pradhan, H.K., Völker, C., Losa, S.N., Bracher, A., Nerger, L., 2019. Assimilation of Global Total Chlorophyll OC-CCI Data and Its Impact on Individual Phytoplankton Fields. J. Geophys. Res. Oceans 124, 470–490. https://doi.org/10.1029/2018JC014329

---

## Author Comment (AC2)

**Review by Anonymus Referee #2**

Review of "Deep chlorophyll maximum and nutricline in the Mediterranean Sea: emerging properties from a multi-platform assimilated biogeochemical model experiment" by Teruzzi et al.

The manuscript addresses the performance of a 3D-Var biogeochemical data assimilation system, constrained with both chlorophyll data from satellite and chlorophyll and nitrate data from BGC-Argo floats, that is applied to a realistic simulation of the Mediterranean Sea for the year 2015. After demonstrating the validity of the method, the authors use their product to investigate the spatial and seasonal variability in the vertical structure of chlorophyll and nitrate fields. The problem is introduced clearly, methodology appears sound, and results are compelling. I recommend the manuscript for publication in Biogeosciences after revisions. Below are a couple of important results that need to be clarified, a list of issues to be addressed, and some minor comments.

**We thank the Anonymous Reviewer for the positive comment on our manuscript and the careful reading. All the points raised by the Reviewer are addressed in the following with replies in green italic.**

1) Fig 2: With the exception of the "LEV" region, it seems that assimilating only float data yields a better fit with chl obs than assimilating floats + satellite. It seems like it would be best to ignore satellite data, even at the surface. On the other hand (Fig 3), assimilating satellite Chl yields a better fit with nitrate obs, which is counter intuitive, and assimilating float Chl barely has an impact. Please comment!

**This Reviewer's comment will help us to clarify some aspects that in the current manuscript version are probably not appropriately presented and discussed:**

In Fig. 2, RMSDs are calculated with respect to float chlorophyll. It is not obvious that satellite chlorophyll assimilation would positively affect metrics with respect to float chlorophyll because of erroneous propagation of surface information along the vertical by DA and because of potential inconsistency between satellite and float data. Indeed, we discussed the inconsistency between satellite and float chlorophyll in the Discussion section, where a comparison between values of chlorophyll concentration from satellite and float highlights their differences. Thus, since satellite and BGC-Argo float inconsistency at surface and because vertical covariance is a prescribed propriety, the change in chlorophyll and BGC-Argo chlorophyll profiles (Fig. 2). On the other hand, considering RMSDs with respect to satellite chlorophyll (Fig.5), the effect of inconsistency between satellite and float chlorophyll assimilation. Thus, we think that the model-assimilation can act as a filter solving inconsistency between sensors, meaning that even if the performance of the multi-platform assimilation is lower than anyone of the single assimilation, it produces a balanced solution with respect to all the available information.

Concerning Fig. 3:

- With the exception of TYR, the assimilation of satellite chlorophyll reduces the RMSD of nitrate (computed on BGC-Argo data) with respect to REF simulation. This is because satellite assimilation correct surface modelled phytoplankton dynamics continuously during late winter/spring (i.e., reducing bloom maxima), and, as a results the entire profile of the phytoplankton is nearly uniformly modified (i.e., reduced) in the euphotic layer. In turn, by reacting to the new phytoplankton conditions, less nitrogen is eventually re-mineralized to nitrate and nitrate concentration is modified in the direction of reducing the REF overestimation in the upper layer. This mechanism has been investigated in previous applications of satellite chlorophyll assimilation (Teruzzi et al., 2018).

- In the case of BGC-Argo chlorophyll assimilation, the changes in the phytoplankton profiles are less uniform (Cossarini et al., 2019), often alternating positive and negative increments along the same profile. The effects on nitrate profiles (through new uptake or release after mortality and exudation) are non-linear and not uniform. It follows that also RMSDs with respect to float nitrate can both decrease or increase with respect to REF. Effects on non-assimilated biogeochemical variables are discussed in a number of works (e.g., Ciavatta et al., 2014; Ford, 2020; Mattern et al., 2017; Santana-Falcón et al., 2020; Simon et al., 2015; Teruzzi et al., 2018; Tsiaras et al., 2017; Yu et al., 2018), where the non-degradation of non-assimilated variables is considered a good result of the assimilation process. In the present manuscript, we briefly discussed the effects of satellite chlorophyll assimilation on nitrate RMSDs (L. 225-228) but we did not summarizes these effects in the Discussion.

Considering the points highlighted above, we will modify the manuscript adding a paragraph dedicated to the effects on non-assimilated variables. In particular, we will focus on the mixed effects of satellite chlorophyll assimilation, which slightly degrade metrics with respect to float chlorophyll but improves those with respect to float nitrate. In perspective, when the inconsistency between satellite and float will be solved, the multiplatform assimilation will provide improvements over large areas thanks to the relevant spatial coverage of satellite observations. In the meantime, as discussed in the manuscript (and reinforced by new comments) the model, acting as a dynamical filter, effectively integrates both sources of information.

2) L. 242-244, 260-261: In Fig 4 I don't see a reduction in RMSD, but instead an increase in levels 4-6. So data assimilation does reduce the model skill in fitting oxygen? It looks like the pink and red curves are on top of each other, suggesting that satellite Chl is responsible for degrading the O2 solution.

Thanks to this Reviewer's comment on the slight degradation of oxygen solution (Fig. 4), we went carefully through the oxygen validation results. Firstly, we considered a new recently updated dataset of BGC-Argo oxygen measurements available at the Coriolis/Ifremer data assembly centre (float trajectories are shown in Fig. R2). Using the updated oxygen dataset, the recalculated RMSD values (Fig. R3) are lower than in Fig. 4 of the original manuscript, indicating that the simulation better compares with respect to the more recent and possibly more reliable observations. At the same time, RMSDs very slightly differ among simulations, especially in the eastern sub-basins. Moreover, the effect of satellite chlorophyll assimilation is not univocal, since RMSDs are both slightly reduced (TYR) and increased (NWM). On the other hand, the float assimilation has a very little effect on oxygen RMSDs. The limited and non-univocal effects of the assimilation on oxygen metrics are related to the interaction of a number of trophic processes (e.g., phytoplankton production and respiration, zooplankton and bacteria respiration) after the assimilation changes on phytoplankton biomass. We think that the non-degradation of the oxygen RMSDs is a good result of the assimilated simulations.

*Fig. R2. Positions of BGC-Argo floats equipped with sensors to provide chlorophyll (blue), nitrate (orange) and oxygen (red) and limits of the subbasins.*

Fig. R3. RMSD between model simulations and BGC-Argo oxygen data in four sub-basins. Grey lines indicate the limits of layers L1-L8 used to calculate the RMSD. The depth scale is different above and below 150 m (double grey line).

We also calculated RMSDs using only oxygen profiles at locations where floats were assimilated (Fig. R4 and R5), thus excluding profiles far from float assimilations. While, oxygen RMSDs are further reduced in this case in the surface layer in NWM and in almost all layers of LEV, differences of RMSDs between simulations are very small also for this dataset and similar to those of Fig. R3.

According to the above considerations, we will update the manuscript introducing metrics based on the updated BGC-Argo oxygen dataset. In particular, Fig. 1 and Fig. 4 will be replaced by Fig. R2 and Fig. R3, respectively. Moreover, we will comment on the slight effect on oxygen in the assimilated simulations, which do not insert degradation on the non-assimilated variable.

---

## Author Response (AR1)

**Deep chlorophyll maximum and nutricline in the Mediterranean Sea: emerging properties from a multi-platform assimilated biogeochemical model experiment**

By Anna Teruzzi, Giorgio Bolzon, Laura Feudale, Gianpiero Cossarini

**Point by point replies to review**

Replies are in italic green, while we report in italic blue parts of the new manuscript version. Line numbers into brackets refer to the new version of the manuscript.

**Review by Sarah Schlunegger**

Referee comments on "Deep chlorophyll maximum and nutricline in the Mediterranean Sea: emerging properties from a multi-platform assimilated biogeochemical model experiment" by Teruzzi et al., currently in discussion at Biogeosciences.

Teruzzi et al presents the results from a data-assimilating biogeochemical model of the Mediterranean Sea. This work provides a novel case-study of the dual use of remotely-sensed (ocean color) and in situ (BGC-Argo) biogeochemical constraints in reconstruction of one year of the biogeochemical state of the Mediterranean Sea. Firstly, the data-informed model solution demonstrates fidelity with observations for a number of surface and depth-resolved biogeochemical metrics, such as the vertical position and longitudinal gradients of the deep chlorophyll maximum and its co-variance with the nutricline. Secondly, the study presents compelling evidence for the synergistic benefits of assimilating estimates of remotely-sensed chlorophyll concentrations in tandem with in situ, depth-resolved estimates of chlorophyll. Thirdly, the study discuses some of the nuanced differences between the impacts of assimilating satellite versus BGC-Argo chlorophyll upon the model solution. The most striking difference discussed is the strong seasonal signatures the different observational streams have upon the solution, with remotely-sensed chlorophyll providing stronger constraint during winter months, and in situ chlorophyll providing stronger constraint during summer months. Finally, implications for optimized sampling strategies, such as the recommendation that BGC-Argo increase sampling frequencies during the 'influential' summer season, are discussed.

This work represents a timely contribution to the underway community efforts towards optimizing the design of a global, autonomous biogeochemical observing network for use in constraining reconstructions of the evolving ocean state. I recommend this manuscript for publication after a few minor clarifications and elaborations are incorporated, as I outline below.

We thank Sarah Schlunegger for the positive and constructive comments on the manuscript. All the points raised in her review are addressed below.

**Points of clarification:**

The model setup describes the biogeochemical model as being coupled offline to the dynamical model. Coupled implies two-directional influence, i.e. that the biogeochemistry is both impacted by and impacts the dynamical fields. Is this the case?

If it is the case that the model set up is "coupled" in the true sense of the word, then the impact on the dynamical fields should also be presented.

If it is not the case, and in fact the biogeochemical model or assimilation of biogeochemical fields does not feed-back or modulate the underlying dynamical fields, then a more appropriate term to use is "forced" or "driven" – i.e. Line 135 should read "the MedBFM was forced or driven offline with output from MENO3.4-OceanVar..."

In MedBFM, the biogeochemical variables are tracers advected and diffused by ocean dynamics and transformed by biogeochemical processes, adopting a source splitting operator (Butenschön et al., 2012) without feed-back on the dynamical fields. In particular, transport and advection are driven by precomputed daily 3D fields of currents and diffusivity, while T and S are used in the biogeochemical module. We modified the manuscript following the Reviewer's suggestion to better explain this aspect as follows:

(L. 142-153) "In the current application, the MedBFM was forced offline with outputs from the NEMO3.2 model of the Mediterranean CMEMS model system (Simoncelli et al., 2016)"

To improve accessibility, an additional paragraph of description or context of Eq. 1 should be included. This would involve, for instance, expalaination of the "innovation" term (Line 159), and discussing the significance or intuitive purpose of the different covariance vectors.

In the new version of the manuscript we revised the description of terms used in eq. (1):

(L. 161-170) "In 3DVarBio, assimilation is performed through the minimization of a cost function that is defined on the basis of Bayes' theorem (Lorenc, 1986) as the weighted sum of the square mismatches between the model background state  $x_b$  (the model state before the assimilation) and the analysis  $x_a$  (the assimilation result) and the observations y. Each square mismatch is weighted according to its accuracy estimations, meaning that  $x_a - x_b$  is weighted by the background error covariance matrix B while (y- H(xb)) by the observation error covariance matrix R:

$$J(x_a) = (x_a - x_b)^T B^{-1} (x_a - x_b) + (y - H(x_b))^T R^{-1} (y - H(x_b)).$$
(1)

In eq. (1)  $\mathbf{y} - H(\mathbf{x}_b)$  is usually named innovation and H the observational operator that maps the values of the model background state  $\mathbf{x}_b$  in the observation space. In our application  $H(\mathbf{x}_b)$  are model values of the variables observed by satellite or floats at observation locations. Through the minimization of the cost function (1), the assimilation provides the analysis  $\mathbf{x}_a$  i.e., the optimal weighted distance from both  $\mathbf{y}$  and  $\mathbf{x}_b$ "

Regarding the "Impact indicator" metric:

(a) It needs to be explained that the impact indicator given by Eq 2, although it does not contain an explicit 'directionality' of the impact (i.e. towards better or worse agreement with the data-constraints), that the assimilation methodology works to push the model solution towards the data, therefore any non-zero value of I(t) represents a nominal improvement in model-data fit. From the equation alone, and without sufficient prior understanding of the methodology, this is not obvious, making it difficult to interpret if the additional streams of assimilated data are merely influencing or in fact improving the solution.

Thank you for the very constructive comments about the Impact Indicator. Concerning the first comment (a), it is true that the impact is in the direction of the assimilated observations for grid-points corresponding to observation locations. On the other hand, grid-points without observations are corrected consistently with the solution of the DA scheme (e.g., the effects of surface chlorophyll assimilation on the water column profiles) with impacts vanishing for grid-points far from any observation locations. Thus, the impact indicator is higher where the assimilation brings the simulation toward the observations and this can be considered an

*improvement of the simulation with respect to nearby assimilated observation. As suggested by the Reviewer, modified the description of the impact indicator in order to highlighting this aspect:*

(L. 286-290) "The subscript 200 represents the integral over the 0-200 m layer, while the overbar represents the average over the whole Mediterranean and over seasonal periods. The impact indicator  $I_{xy}(t)$  was calculated for each assimilation date and each grid point, and then, statistically analysed and summarized on a seasonal base. The indicator  $I_{xy}(t)$  quantifies how much an assimilated run deviates from the REF simulation, thus it is higher where and when the simulation deviates from REF, and is closer than REF to the assimilated observations."

Moreover, to clarify that impact indicator is calculated for each model grid-point, we added the xy subscript in its definition and in all the occurrences:

$$(L. 284) "I_{xy}(t) = \frac{|ScFcn(t) - REF(t)|_{200}}{\overline{REF(t)}_{200}}$$
(2)"

(b) The maps of Fig. 6 and Fig. 7 present a very derived / abstract metric. These figures present the 50th and 90th percentile 'impact' of assimilated the given field during the given season. Interpretability of this abstracted metric would improve if an additional subpanel was included in each figure which showed a representative distribution of the impact-indicator for a single grid-cell (or averaged over a region), with markings at the 50th and 90th percentile. This could also provide opportunity to contrast the summer vs. winter distributions of the indicator. For instance, for chlorophyll, the winter distribution of the indicator would be shifted toward "1" while the summer distribution would be closer to "zero". See a mock-up below. This will help orient the reader as to what the maps are presenting.

We thank the Reviewer for this constructive comment about the Impact Indicator. Following the Reviewer's suggestion, we investigated the distribution of the Impact Indicator over winter and summer. In particular, we analysed frequency distributions of the impact indicator at some locations, and we provide in Fig. R1 two examples. The histograms confirm results shown in Fig. 6 and 7, with higher Impact factor percentiles in winter with respect to summer for chlorophyll and the opposite for nitrate (since the impact of sparse float observations is negligible at the simulation beginning). Further, the histograms are quite scattered since they depend on a number of factors (e.g., the number of observations in the area and assimilation changes during subsequent dates). Therefore, we think that the non-parametric statistics (percentiles) shown in the manuscript are a sound choice to quantify the impact of the assimilation on model solution.

Fig. R1. Frequency histograms of the Impact Indicator at two locations for chlorophyll (top) and nitrate (bottom) in winter (blue) and summer (orange). Vertical lines indicates the values of the 50th (p50, dashed line) and 95th (p95, continuous line) percentiles.

The conclusions section would benefit from a final sentence that poses the significance of the study within the context of future advances in biogeochemical data assimilation within basin and global domains, i.e. something like "The multi-platform assimilation yielded improvements in model representation of large-scale (hundreds to thousands of kilometers) bio-dynamical features and is suggestive of the applicability of this advancement to reconstructions of other ocean regions and the global domain."

We thank the reviewer for this suggestion that helps to enlarge the perspective of our manuscript. We added a comment about the applicability of multi-platform DA to other domains: :

(L. 517-521) "The impacts of multi-variate profile assimilation are directly linked to the sampling frequency and dimension of the BGC-Argo network, which should increase to match the consolidated importance and relevance of satellite observation assimilation. Thus, in a perspective view, the multi-platform assimilation can improve model representation of both large-scale (hundreds to thousands of kilometres) to mesoscale features and be beneficial for robust reconstruction in global and regional reanalysis."

**Minor editorial and stylistic suggestions:**

L7-10: Rewrite to something like: "Data assimilation has lead to advancements in biogeochemical modelling and scientific understanding of the ocean. The recent operational availability of data from BGC-Argo floats, which provide valuable insights into key vertical biogeochemical processes, stands to further improve biogeochemical modelling through assimilation schemes that include observations from floats in addition to traditionally assimilated satellite data." (bold is new)

Thank you for the suggestion. We included it in the revised manuscript (L. 7-9).

L16: "maximum depth, intensity and nutricline depth" (added a comma and removed the first 'and')

**We modified the manuscript accordingly (L. 17).**

L39: Add the following reference, which also provides motivaiton for the direct use of optical properties in data assimilation:

Dutkiewicz, Stephanie, Anna E. Hickman, Oliver Jahn, Stephanie Henson, Claudie Beaulieu, and Erwan Monier. 2019. "Ocean Colour Signature of Climate Change." Nature Communications. 10 (1). https://doi.org/10.1038/s41467-019-08457-x.

**We inserted the reference (L. 38)**

L40: Ocean-colour observation assimilation takes advantage of the frequent, large-scale satellite observations of ocean properties related to the microbial biology of the upper ocean.

L42: "deeper ocean layers requires approximations and assumptions."

We changed the manuscript according to the two previous Reviewer's suggestions (L. 40-41 and L. 42).

**L45: "localization" - what does this mean?**

In Ensemble Kalman filter (EnKF)-like data assimilation, "localization" is a method applied to limit the impact of the assimilation (increments) to areas relatively close to the observations. In particular, localization is applied when knowledge of spatial covariance is relevantly approximated due to the limited ensemble size. In this sense, localization avoid to insert spurious and non-realistic effects and can be applied not only to limit the spatial covariance but also on covariance among variables or on the time dimension. Concerning the use of "localization" in the sentence at L. 45, we are referring to vertical localization applied in some DA application of satellite ocean colour, meaning that the assimilation impacts are limited to a portion of the water column through localization. We modified the sentence as follows to make it more clear about "localization" in EnKF-like data assimilation:

(L. 42-46) "Vertical covariance must be parameterized by synthetic precalculated vertical profiles in variational schemes (Teruzzi et al., 2018), while EnKF-like (ensemble Kalman filter) schemes may have limitations in effectively impacting deeper ocean layers (Fontana et al., 2013; Hu et al., 2012). Indeed, some EnKF-like applications introduce limitation to the increments in subsurface layers through localization in the vertical direction to address spurious correlations (Goodliff et al., 2019; Pradhan et al., 2019)."

L66: "ideal" – replace with "suitable" or "appropriate" – as it could be easily argued that a region with less coastal margins and more open-ocean conditons, where ocean color is more reliable, would be a more 'ideal' location to do a multi-platform assimilation.

We replaced "ideal" with "suitable" (L. 67).

L348: "In our results, this hypothesis was supported by higher nutrient uptakes in the western..."

We replaced "scenario" with "hypothesis" (L. 364).

L409: Remind readers of what Vv means, i.e. "Vv (vertical co-varience error)"

L415: Remind readers of what Vb means, i.e. "Vb (biogeochemical co-varience error)"

*In the reviewed Discussion the meaning of the operators is reminded at the first occurrence of each operator (L. 429 and L. 459).*

**Review by Anonymus Referee #2**

Review of "Deep chlorophyll maximum and nutricline in the Mediterranean Sea: emerging properties from a multi-platform assimilated biogeochemical model experiment" by Teruzzi et al.

The manuscript addresses the performance of a 3D-Var biogeochemical data assimilation system, constrained with both chlorophyll data from satellite and chlorophyll and nitrate data from BGC-Argo floats, that is applied to a realistic simulation of the Mediterranean Sea for the year 2015. After demonstrating the validity of the method, the authors use their product to investigate the spatial and seasonal variability in the vertical structure of chlorophyll and nitrate fields. The problem is introduced clearly, methodology appears sound, and results are compelling. I recommend the manuscript for publication in Biogeosciences after revisions. Below are a couple of important results that need to be clarified, a list of issues to be addressed, and some minor comments.

**We thank the Anonymous Reviewer for the positive comment on our manuscript and the careful reading. All the points raised by the Reviewer are addressed in the following.**

1) Fig 2: With the exception of the "LEV" region, it seems that assimilating only float data yields a better fit with chl obs than assimilating floats + satellite. It seems like it would be best to ignore satellite data, even at the surface. On the other hand (Fig 3), assimilating satellite Chl yields a better fit with nitrate obs, which is counter intuitive, and assimilating float Chl barely has an impact. Please comment!

**This Reviewer's comment helped us to clarify some aspects that in the current manuscript version are not probably appropriately presented and discussed:**

In Fig. 2, RMSDs are calculated with respect to float chlorophyll. It is not obvious that satellite chlorophyll assimilation would positively affect metrics with respect to float chlorophyll because of erroneous propagation of surface information along the vertical and because of potential inconsistency between satellite and float data. Indeed, we discussed the inconsistency between satellite and float chlorophyll in the Discussion section, where a comparison between values of chlorophyll concentration from satellite and float highlights their differences. Thus, since satellite and BGC-Argo float inconsistency at surface and because vertical covariance is a prescribed propriety, the change in chlorophyll and BGC-Argo chlorophyll profiles (Fig. 2). On the other hand, considering RMSDs with respect to satellite chlorophyll (Fig.5), the effect of inconsistency between satellite and float chlorophyll assimilation. Thus, we think that the model-assimilation can act as a filter solving inconsistency between sensors, meaning that even if the performance of the multi-platform assimilation is lower than anyone of the single assimilation, it produces a balanced solution with respect all the available information.

Concerning Fig. 3:

- With the exception of TYR, the assimilation of satellite chlorophyll reduces the RMSD of nitrate (computed on BGC-Argo data) with respect to REF simulation. This is because satellite assimilation correct surface modelled phytoplankton dynamics continuously during late winter/spring (i.e., reducing bloom maxima), and, as a results the entire profile of the phytoplankton is nearly uniformly modified (i.e., reduced) in the euphotic layer. In turn, by reacting to the new phytoplankton conditions, less nitrogen is eventually re-mineralized to nitrate and nitrate concentration is modified in the direction of reducing the REF overestimation in the upper layer. This mechanism has been investigated in previous applications of satellite chlorophyll assimilation (Teruzzi et al., 2018, 2014).

- In the case of BGC-Argo chlorophyll assimilation, the changes in the phytoplankton profiles are less uniform (Cossarini et al., 2019), often alternating positive and negative increments along the same profile. The effects on nitrate profiles (through new uptake or release after mortality and exudation) are non-linear and not uniform. It follows that also RMSDs with respect to float nitrate can both decrease or increase with respect to REF. Effects on non-assimilated biogeochemical variables are discussed in a number of works (e.g., Ciavatta et al., 2014; Ford, 2020; Mattern et al., 2017; Santana-Falcón et al., 2020; Simon et al., 2015; Teruzzi et al., 2018; Tsiaras et al., 2017; Yu et al., 2018), where the non-degradation of non-assimilated variables is considered a good result of the assimilation process. In the present manuscript, we briefly discussed the effects of satellite chlorophyll assimilation on nitrate RMSDs (L. 225-228) but we did not summarized these effects in the Discussion.

Considering the points highlighted above, we modified the manuscript adding paragraphs dedicated to the effects on non-assimilated variables. In particular, we focused on the mixed effects of satellite chlorophyll assimilation, which slightly degrade metrics with respect to float chlorophyll but improves those with respect to float nitrate. In perspective, when the inconsistency between satellite and float will be solved, the multiplatform assimilation will provide improvements over large areas thanks to the relevant spatial coverage of satellite observations. In the meantime, as discussed in the manuscript (and reinforced by new comments) the model, acting as a dynamical filter, effectively integrates both sources of information. Hereafter the new paragraphs added in the manuscript:

(L. 422-431) "As a consequence of these discrepancies and of the propagation of information through the prescribed DA vertical covariance , the change in chlorophyll profiles due to satellite assimilation either reduced and increased the distance between modelled chlorophyll and BGC-Argo chlorophyll profiles (Fig. 2). On the other hand, considering RMSDs with respect to satellite chlorophyll (Fig. 5), the effect of inconsistency between satellite and float chlorophyll was highlighted by the slight increase in RMSDs in simulations with float chlorophyll assimilation. In perspective, when the inconsistency between satellite and float will be solved, the multi-platform assimilation will provide improvements over large areas thanks to the relevant spatial coverage of satellite observations. In the meantime, the model, acting as a dynamical filter, integrates both sources of information together with the vertical covariance V\_V operator implemented in the assimilation scheme. Indeed, the vertical covariance is a key element that allows to integrate surface with sub-surface information."

(L. 439-449) "Considering effects on nitrate, the assimilation of satellite chlorophyll reduced the nitrate RMSDs (computed on BGC-Argo data) with respect to REF simulation in all the sub-basins with exception of TYR (Fig. 3). The rather persistent and broad DA increments during late winter and early spring were acting to reduce overestimation of the bloom maxima, resulting in nearly uniform reductions of the phytoplankton biomass in the euphotic layer. In turn, by reacting to the new phytoplankton concentration, less nitrogen was eventually re-mineralized to nitrate. Thus, the nitrate concentration was modified in the direction of reducing the REF nitrate overestimation in the upper layer. An analogous mechanism has been investigated in previous applications of satellite chlorophyll assimilation in the MedBFM system (Teruzzi et al., 2018, 2014). In the case of BGC-Argo chlorophyll assimilation, the changes in the phytoplankton profiles were less uniform (Cossarini et al., 2019), often alternating positive and negative increments along the same profile. It follows that the effects on nitrate profiles (through new uptake or release after mortality and exudation) were non-linear and not uniform, with relatively small impacts on the overall nitrate RMSDs."

2) L. 242-244, 260-261: In Fig 4 I don't see a reduction in RMSD, but instead an increase in levels 4-6. So data assimilation does reduce the model skill in fitting oxygen? It looks like the pink and red curves are on top of each other, suggesting that satellite Chl is responsible for degrading the O2 solution.

Thanks to this Reviewer's comment on the slight degradation of oxygen solution (Fig. 4), we went carefully through the oxygen validation results. Firstly, we considered a new recently updated dataset of BGC-Argo

oxygen measurements available at the Coriolis/Ifremer data assembly centre (float trajectories are shown in Fig. R2). Using the updated oxygen dataset, the recalculated RMSD values (Fig. R3) are lower than in Fig. 4 of the original manuscript, indicating that the simulation better compares with respect to the more recent and possibly more reliable observations. At the same time, RMSDs very slightly differ among simulations, especially in the eastern sub-basins. Moreover, the effect of satellite chlorophyll assimilation is not univocal, since RMSDs are both slightly reduced (TYR) and increased (NWM). On the other hand, the float assimilation has a very little effect on oxygen RMSDs. The limited and non-univocal effects of the assimilation on oxygen metrics are related to the interaction of a number of trophic processes (e.g., phytoplankton production and respiration, zooplankton and bacteria respiration) after the assimilation changes on phytoplankton biomass. We think that the non-degradation of the oxygen RMSDs is a good result of the assimilated simulations.

---

## Author Response (AR2)

**Deep chlorophyll maximum and nutricline in the Mediterranean Sea: emerging properties from a multi-platform assimilated biogeochemical model experiment**

By Anna Teruzzi, Giorgio Bolzon, Laura Feudale, Gianpiero Cossarini

**Reply to Reviewer's comments**

The authors have thoroughly addressed my comments. Here are some minor edits:

line 76 "one year(2015)": space missing

line 284 refers to $|S(t)-R(t)|$ but equation 2 is written as $|ScFcn(t)-REF(t)|$

line 287 "analysed" -> analyzed

line 292 "where at least one assimilation remarkably ameliorated the model mismatch" -> where at least one assimilated observation remarkably reduced the model mismatch

line 424: "either reduced and increased" -> either reduced or increased

*We thank the Reviewer for her/his positive feedback. We addressed all the minor edits in the final version of the manuscript.*